# Structure-Aware Cooperative Ensemble Evolutionary Optimization on Combinatorial Problems with Multimodal Large Language Models

**Jie Zhao[1], Kang Hao Cheong[1,2]***

[1]School of Physical and Mathematical Sciences, Nanyang Technological University
[2]College of Computing and Data Science, Nanyang Technological University

## Abstract

Evolutionary algorithms (EAs) have proven effective in exploring the vast solution spaces typical of graph-structured combinatorial problems. However, traditional encoding schemes, such as binary or numerical representations, often fail to straightforwardly capture the intricate structural properties of networks. Through employing the image-based encoding to preserve topological context, this study utilizes multimodal large language models (MLLMs) as evolutionary operators to facilitate structure-aware optimization over graph data. To address the visual clutter inherent in large-scale network visualizations, we leverage graph sparsification techniques to simplify structures while maintaining essential structural features. To further improve robustness and mitigate bias from different sparsification views, we propose a cooperative evolutionary optimization framework that facilitates cross-domain knowledge transfer and unifies multiple sparsified variants of diverse structures. Additionally, recognizing the sensitivity of MLLMs to network layout, we introduce an ensemble strategy that aggregates outputs from various layout configurations through consensus voting. Finally, experiments on real-world networks through various tasks demonstrate that our approach improves both the quality and reliability of solutions in MLLM-driven evolutionary optimization.

## 1 Introduction

Graph-structured combinatorial problems in real-world complex systems occupy a central role in various domains [1]. These problems require selecting an optimal set of nodes within a network and are inherently NP-hard due to the combinatorial explosion of possible node combinations [2, 3]. Evolutionary algorithms (EAs) have been extensively employed to address such challenges because of their ability to effectively deal with non-linearity and explore discrete search spaces [4].

The encoding scheme plays a fundamental role in addressing discrete optimization problems, as it dictates how solutions are structured and operated within the EA framework [5, 6]. Different combinatorial tasks have unique requirements, for example, permutation and path encodings are commonly used for scheduling [7] and routing problems [8], while binary and index-based encodings are often applied to node or edge subset selection tasks [9]. Despite their widespread use, conventional encoding schemes represent data in an abstract way, overlooking their contextual significance within the network. Therefore, evolutionary operators like crossover and mutation are applied blindly, without recognizing the underlying structural relationships of solutions.

---

*Corresponding author: `kanghao.cheong@ntu.edu.sg`

39th Conference on Neural Information Processing Systems (NeurIPS 2025).

An image-based encoding framework, combined with multimodal large language models (MLLMs) as evolutionary operators, offers a practical approach for complex network optimization [10]. Visual representations can preserve structural and contextual nuances that are often lost in string-based encodings. By virtue of their capacity to interpret both textual and visual information [11], MLLMs facilitate the integration of context awareness into the optimization process, rendering them especially effective for tackling combinatorial problems in networks.

However, visualizing large-scale networks often leads to cluttered plots, hindering structural interpretation. Graph sparsification can help reduce this complexity while preserving key topological features [12, 13]. Yet, any single sparsified view may introduce bias by overemphasizing specific structures. Therefore, we distribute the optimization process across multiple sparsified versions, allowing diverse structural insights to emerge and reducing reliance on any single simplification.

While cooperative evolutionary optimization has shown success in handling complex tasks [14, 15], traditional approaches generally rely on the assumption of a shared domain, which is incompatible with our multi-sparsification setting. To make the cooperative framework adaptable to the multi-domain case, we implement a master-worker architecture, where a central coordinator directs subprocessors operating on distinct sparsified networks. Moreover, we establish a cross-domain mapping mechanism to enable effective knowledge transfer across these subprocessors.

Recent studies indicate that MLLMs are sensitive to variations in network layouts [16], which can influence optimization outcomes. To address this limitation, we propose an ensemble model that integrates multiple layout styles. By employing a consensus voting mechanism to fuse their outputs, we are able to enhance robustness and mitigate layout-induced bias. To demonstrate the effectiveness of the proposed method, an important combinatorial problem, influence maximization, is utilized as the main case study [17, 18]. Our key contributions are summarized as follows:

- To counteract the bias of relying on any single simplified network, we distribute the optimization process across multiple sparsified versions. This approach preserves diverse structural insights and improves optimization performance.

- In the cooperative framework, we adopt the master-worker model to coordinate optimization across heterogeneous sparsified networks. By establishing the cross-domain mapping mechanism, knowledge is effectively transferred among subprocessors operating on networks of different scales and characteristics.

- Recognizing the sensitivity of MLLMs to network visualization layouts, we develop an ensemble model that incorporates multiple layouts. A consensus voting mechanism is then proposed to aggregate these representations to mitigate layout-induced variance and improve the optimization robustness.

**Illustrative task:** The focus of this paper is not on advancing the state-of-the-art in the influence maximization problem, but rather on using this challenging problem as a representative task to demonstrate how MLLMs can function as structure-aware evolutionary optimization operators. Moreover, several additional toy tasks are presented in Section 4.4 to demonstrate the generalizability and potential of structure-aware optimization. The implementation code is available online at `Structure-Aware EO`.

## 2    Related Work

### 2.1    Evolutionary Optimization on Combinatorial Problems

EAs have proven effective across various combinatorial tasks, including routing [8], scheduling [19], network robustness enhancement [20] and network reconstruction [21]. Their adaptability also extends to tasks like community deception [22], sensor selection [23] and dynamic community detection [24]. In EAs, the encoding strategies are problem-specific, e.g., permutation/path encodings for scheduling and routing [25], and binary/label-based encoding for influence maximization [26]. However, such encoding schemes often lack structural awareness, leading to naive modifications.

## 2.2 LLM-Assisted Evolutionary Optimization

Recent work has integrated LLMs into evolutionary frameworks [27, 28], particularly as search operators [29, 30] and automation tools [31]. Specifically, LLMs could be used to implement mutation and crossover [32, 33], and support multi-objective optimization [34, 35]. However, LLMs struggle to capture higher-order structural information, and representing networks as natural language will incur substantial computational costs, making them less suitable for structure-aware optimization. An alternative is offered by MLLMs with image-based encodings, which can address such problems intuitively in a human-like manner [10].

## 3 Cooperative and Ensemble MLLM-based Evolutionary Optimization

In this section, we provide a detailed explanation of the proposed framework, including graph sparsification and cooperative/ensemble MLLM-based evolutionary optimization.

### 3.1 Problem Formulation

Given a network $G = (V, E)$, the goal of the influence maximization problem is to select a seed set $S \subseteq V$ of size $k$ that maximizes the expected influence spread:

$$\max_{S \subseteq V, |S|=k} \sigma(S),$$

where $\sigma(S)$ denotes the expected number of nodes activated under a diffusion model. Estimating $\sigma(S)$ typically requires costly Monte Carlo simulations, posing scalability issues. Therefore, we adopt the expected diffusion value (EDV) [36] as a surrogate:

$$\textbf{EDV}(S) = k + \sum_{b \in \mathcal{N}(S) \setminus S} \left( 1 - (1-p)^{\delta(b)} \right),$$

where $p$ is the influence probability, $\delta(b)$ counts the edges from $S$ to $b$, and $\mathcal{N}(S)$ is the one-hop neighborhood of $S$, including $S$ itself.

### 3.2 Graph Sparsification

Visualizing large-scale labeled networks can result in excessive clutter, affecting MLLMs' recognition on graph structure. To address this, we apply graph sparsification to obtain a miniature version of the original networks that preserves essential structures while reducing noise. Let the original network be $G = (V, E)$; a sparsified version is produced via a sparsification operator $\mathcal{S}(\cdot; \theta)$, yielding:

$$G_s^i = \mathcal{S}(G; \theta_i),$$

where $\theta_i$ is the sparsification strategy. Different sparsification methods preserve different structural properties, with some emphasizing global connectivity and others focusing on local clustering. By applying multiple strategies, we can generate a set of simplified networks capturing diverse aspects of the original:

$$G_s = \{G_s^1, G_s^2, \ldots, G_s^n\}, \quad \text{where } G_s^i \subset G.$$

In this work, we employ two heuristic graph sparsification techniques to reduce network size while preserving essential structural properties from various perspectives.

#### 3.2.1 Degree-based Sparsification

In this method, we simplify the graph by retaining nodes with high degree. Given a graph $G = (V, E)$, we define $N_v^*$ as the number of nodes to retain, and the subset $V_s$ of retained nodes is:

$$V_s = \{v_i \in V \mid d(v_i) \geq d(v_j), \forall v_j \in V \setminus V_s\} \text{ with } |V_s| = N_v^*,$$

where $d(v)$ represents the degree of the node $v$.

### 3.2.2 Community-based Sparsification

Given a partition of $G$ into communities $\widetilde{\mathcal{C}} = \{\mathcal{C}_1, \mathcal{C}_2, \ldots, \mathcal{C}_m\}$, the number of selected nodes of each community is determined by:

$$|\mathcal{C}_i'| = \left\lfloor \frac{|\mathcal{C}_i|}{|V|} \cdot N_v^* \right\rfloor.$$

Within each community, the nodes are chosen based on their betweenness centrality values, denoted as $b(v)$ for $v \in V$. The selected nodes $V_s$ are given by:

$$V_s = \bigcup_{i=1}^{m} \{v_j \in \mathcal{C}_i \mid b(v_j) \geq b(v_l), \forall v_l \in \mathcal{C}_i \setminus V_s\} \text{ with } |V_s \cap \mathcal{C}_i| = |\mathcal{C}_i'|.$$

In the community-based sparsification method, the initial community distribution of network is obtained by the FastGreedy algorithm [37].

### 3.2.3 Subgraph Refinement

After selecting the nodes, we construct the subgraph $G_s = (V_s, E_s)$, where

$$E_s = \{(u, v) \in E \mid u, v \in V_s\}.$$

If the number of edges in $E_s$ exceeds the predefined edge number $N_e^*$, we prune edges randomly until only $N_e^*$ remain. Finally, we remove the isolated nodes and retain only the largest connected component to ensure structural coherence.

### 3.3 Knowledge Transfer-based Cooperative Evolutionary Optimization

In our cooperative optimization, each processor optimizes over different sparsified networks $G_s^i = (V_s^i, E_s^i)$ with the knowledge transfer from each other (see Figure 1 for the illustration), optimizing candidate solutions $s \subset V_s^i$. During the optimization, each processor evolves a population $\mathcal{P}_i$, where

$$\forall s \in \mathcal{P}_i, \quad s \in V_s^i.$$

The goal of the $i$-th processor is to find the individual $s_i^*$ with the best performance:

$$s_i^* = \arg\max_{s \in \mathcal{P}_i} f(\phi_i(s); G),$$

where $f(s; G)$ is the fitness function to evaluate solution $s$ regarding the original network $G$. Every candidate solution optimized in $G_s^i \in G_s$ will be mapped back to the domain of network $G$ via function $\phi_i$ for fitness evaluation to ensure that the solutions are relevant and effective with respect to the full network. Following that, the elite solution optimized in one sparsified domain will be transferred into another sparsified domain via the original domain. Specifically, we define two mapping functions:

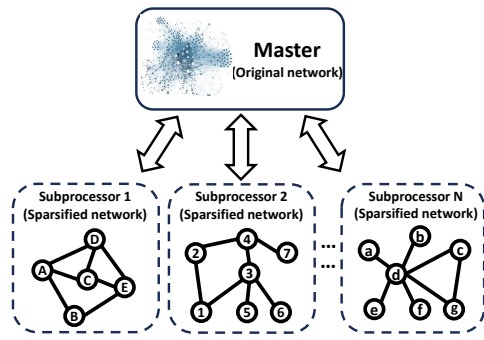

Figure 1: Diagram of the cooperative evolutionary optimization. The master unit stores the elite solutions and is responsible for knowledge transfer across subprocessors.

- **Projection to Original:** A function $\phi_i : V_s^i \to V$ maps nodes from a sparsified network $G_s^i$ to their corresponding nodes in $G$. A candidate solution $s \subset V_s^i$ is projected to the domain of the original network as $\phi_i(s) = \{ \phi_i(v) : v \in s \}$ where $\phi_i(s) \in V$.

- **Injection (Reverse Projection):** A function $\phi_i^- : V \to V_s^i$ maps candidate solutions $s \subset V$ from the original network into the domain of a given sparsified network $G_s^i$, i.e., $\phi_i^-(s) = \{ \phi_i^-(v) : v \in s \}$ where $\phi_i^-(s) \in V_s^i$.

**Remark 1.** *Let $G_s^i$ and $G_s^j$ be two simplified representations of an original network $G$, obtained via distinct graph sparsification methods. Although $\phi_i(v)$ is a valid node in $G$ for any $v \in V_s^i$, there is no guarantee that $\phi_i(v) \in V$ is contained within the vertex set $V_s^j$ of $G_s^j$ when attempting to map it via the injection function $\phi_j^-$. This potential infeasibility arises because different sparsification methods may retain distinct subsets of nodes, even though both sparsified networks originate from $G$.*

The master process coordinates the search by collecting high-quality solutions from the subprocessors. Each subprocessor periodically sends its best candidate (projected via $\phi$) to the master, which aggregates them into a global elite pool $\mathcal{E}$. Specifically, if the $i$-th subprocessor provides its best candidate $s_i^*$ from its sparsified network, the corresponding candidate in the original domain is:

$$S_i = \phi_i(s_i^*),$$

where the mapped candidate $S_i$ will be added into the elite pool $\mathcal{E}$.

### 3.3.1 Update of Master

In our cooperative evolutionary optimization framework, each candidate stored in the global elite pool $\mathcal{E}$ is represented as $(S, f(S), g)$, i.e.,

$$\mathcal{E} = \{(S_{i,j}, f(S_{i,j}), g_j) \mid i = 1, 2, \ldots, n\},$$

where $S_{i,j}$ is the best solution in the $i$-th subprocessor at the $j$-th generation. $f(S)$ is its corresponding fitness (e.g., estimated influence spread), and $g$ is the generation number at which the candidate was added (serving as a timestamp). Then, we can determine the current generation $g_{\text{current}}$ by taking the maximum generation value in $\mathcal{E}$:

$$g_{\text{current}} = \max\{\, g_j \mid (S_{i,j}, f(S_{i,j}), g_j) \in \mathcal{E} \,\}.$$

Then, a threshold parameter $\Delta g$ is defined to specify the maximum allowed age (in terms of generations) for candidates to remain in the pool. The elite pool $\mathcal{E}$ is pruned as follows:

$$\mathcal{E} \leftarrow \{\, (S_{i,j}, f(S_{i,j}), g_j) \in \mathcal{E} \mid g_{\text{current}} - g_j \leq \Delta g \,\},$$

which ensures that only those candidates whose age is at most $\Delta g$ are retained.

Note that $\mathcal{E}$ is sorted in descending order based solely on fitness. Specifically, for any two candidates $(S_{i,j}, f(S_{i,j}), g_j)$ and $(S_{i',j'}, f(S_{i',j'}), g_{j'})$ in $\mathcal{E}$, if $(S_{i,j}, f(S_{i,j}), g_j)$ appears before $(S_{i',j'}, f(S_{i',j'}), g_{j'})$ in the sorted order, then $f(S_{i,j}) \geq f(S_{i',j'})$. Finally, to prevent unbounded growth, the pool $\mathcal{E}$ is limited to size $N_{\mathcal{E}}$, ensuring that the global elite pool focuses on recent and high-quality solutions.

### 3.3.2 Update of Subprocessor

When the $i$-th subprocessor requests an elite candidate, the master may send a candidate $S_j \in \mathcal{E}, i \neq j$ expressed in the original domain. Because the sparsified networks differ in size, $S_j$ may not directly map onto the domain of $G_s^i \in G_s$. To address this issue, we define a projection mechanism for cross-domain injection. Given a solution $S \in \mathcal{E}$ and a sparsified network $G_s^i$, let

$$\phi_i^-(S) = \{\phi_i^-(v) \in V_s^i \mid v \in S, \, S \subseteq V\},$$

where $\phi_i^-(v)$ maps the node $v \in V$ into the domain of sparsified network $G_s^i$. The projected candidate is then obtained by

$$\text{Proj}(S, G_s^i) = \begin{cases} \phi_i^-(S), & \text{if } |\phi_i^-(S)| = k, \\ \phi_i^-(S) \cup \text{Fill}(S, G_s^i), & \text{if } |\phi_i^-(S)| < k, \end{cases}$$

where $k$ is the predefined size of solutions. $\text{Fill}(S, G_s^i)$ (strategically or randomly) selects additional nodes from $V_s^i$ to ensure that the final candidate contains exactly $k$ nodes. This mechanism guarantees that elite solutions from the master can be feasibly injected into any subprocessor's local search space, despite differences among the sparsified networks.

A shared global elite pool $\mathcal{E}$ is maintained by all processors. Every $T$ generations (the elite injection interval), each processor sends its best candidate (after projection to the original network) into $\mathcal{E}$. Conversely, a candidate from $\mathcal{E}$ is injected into the local population of a subprocessor after projection into the sparsified network whenever that candidate performs better than the weakest solution currently in the local population.

## 3.4 Ensemble MLLM-based Evolutionary Optimization

Here, we introduce an ensemble framework leveraging MLLMs for evolutionary optimization on graph-structured data by incorporating multiple visual layouts.

### 3.4.1 Visualization

Since each graph layout imposes distinct structural biases [16], the performance of MLLMs is inherently sensitive to layout choice, and reliance on a single layout risks introducing perceptual distortions. To mitigate this issue, we adopt an ensemble strategy that integrates multiple layouts, providing diverse structural views and enhancing overall performance (see Figure 2).

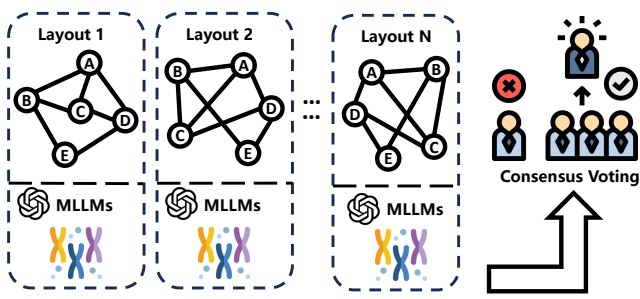

Figure 2: Diagram of the ensemble strategy. Each layout is fed into MLLMs separately, and the final outcome is determined using the consensus voting strategy.

To enable seamless visual incorporation into our evolutionary framework, we use two separate functions designed for generating image representations [10]. One function handles the initialization phase, while the other encodes individual candidate solutions as images for their participation in genetic operations. The MLLM-based initialization is defined as follows:

$$\mathcal{I}_{\text{init}}(G, \Theta) = \textbf{Visualize}(G, \Theta), \tag{1}$$

which renders a visual depiction of graph $G$ based on the layout parameters $\Theta$, utilizing external visualization tools. In contrast, to encode individual candidate solutions in image form, we have

$$\mathcal{I}(G, \Theta, s) = \textbf{Visualize}(G, \Theta, s), \tag{2}$$

which produces a visual representation of graph $G$ arranged according to layout $\Theta$, with the nodes belonging to solution $s$ distinctly marked in color, as illustrated in Figure 3. Note that in practice, during crossover, only the seed nodes are visualized to reduce inference difficulty, whereas in mutation all nodes are visualized.

### 3.4.2 MLLM-based Initialization

Given a graph $G$, an initialization strategy and layout style $\Theta$, the initial population $\mathcal{P}$ is generated as:

$$\mathcal{P} = \textbf{MLLM\_init}(\mathcal{I}_{\text{init}}(G, \Theta); N_p), \tag{3}$$

where $N_p$ denotes the population size.

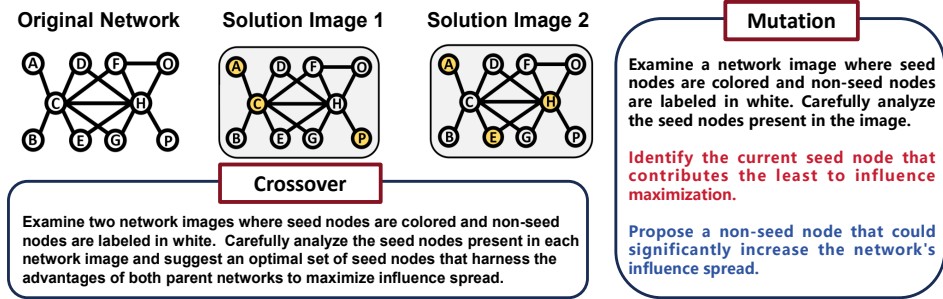

Figure 3: Diagram of the MLLM-based evolutionary optimization on influence maximization. The solution and its underlying network are jointly represented in the image format.

### 3.4.3 Ensemble MLLM-based Crossover and Mutation

Given two candidate solutions $s_i, s_j \in \mathcal{P}$ where $\mathcal{P}$ denotes the population, the crossover operation guided by MLLMs generates an offspring as:

$$s_c = \textbf{MLLM\_crossover}\big(\mathcal{I}(G, \Theta, s_i), \mathcal{I}(G, \Theta, s_j); \mathbb{P}_c\big),$$

where $\mathbb{P}_c$ denotes the crossover probability. To mitigate layout-specific bias, we propose an ensemble strategy that aggregates decisions across multiple layouts $\{\Theta_l\}_{l=1}^{L}$ via:

$$s'_c = \textbf{ConsensusVoting}\left(\{\textbf{MLLM\_crossover}(\mathcal{I}(G, \Theta_l, s_i), \mathcal{I}(G, \Theta_l, s_j))\}_{l=1}^{L}\right),$$

where **ConsensusVoting** aggregates candidate solutions by evaluating the support each node receives in different layouts. Each node accumulates votes according to its frequency of occurrence, and nodes that meet a predefined threshold are chosen first. If the number of qualified nodes is insufficient, additional nodes are selected greedily based on their influence in the graph, measured by the number of uncovered neighbors.

Similarly, the ensemble mutation is defined as:

$$s_m = \textbf{MLLM\_mutate}\big(\mathcal{I}(G, \Theta, s); \mathbb{P}_m\big),$$

$$s'_m = \textbf{ConsensusVoting}\left(\{\textbf{MLLM\_mutate}(\mathcal{I}(G, \Theta_l, s))\}_{l=1}^{L}\right),$$

where $\mathbb{P}_m$ denotes the mutation probability.

## 4 Experimental Study

The experiments are conducted on eight real-world networks, as shown in Table 1. The number of nodes and edges in the simplified networks is fixed at 50 and 100, respectively, to remain manageable for MLLMs. The probabilities of crossover and mutation are set to 0.2 and 0.1. The number of individuals in the initialized population is set to 20. The reported results are averaged from 10 independent simulations unless otherwise specified. The backbone MLLM is *gpt-4o-2024-11-20*.

Table 1: Structural details about the networks: $|V|$ is the number of nodes; $|E|$, the number of edges.

| Network | USAir | Netscience | Polblogs | Facebook | WikiVote | Rutgers89 | MSU24 | Texas84 |
|---|---|---|---|---|---|---|---|---|
| $|V|$ | 332 | 379 | 1,222 | 4,039 | 7,066 | 24,568 | 32,361 | 36,365 |
| $|E|$ | 2,126 | 914 | 16,717 | 88,234 | 100,736 | 784,596 | 1,118,767 | 1,590,651 |

### 4.1 Effectiveness Examination of Cooperative Evolutionary Optimization

In this section, we compare the proposed cooperative optimization (multi-domain) with other evolutionary frameworks (single-domain). Spars-D-EO and Spars-C-EO apply the vanilla evolutionary optimization to degree-based and community-based sparsified networks without involving any advanced techniques, respectively. Co-EO combines both sparsification strategies with a knowledge transfer mechanism. As shown in Table 2, Co-MLLM(KK) integrating MLLM with the KK layout and cooperative optimization achieves the highest fitness values across most networks. For fairness, SAEP [22], CoeCo [38], and SSR [39] optimize the population over the degree-based sparsified network. These methods highlight the advantages of self-adaptive control, divide-and-conquer decomposition, and search-space reduction in addressing complex network optimization tasks. The results indicate the potential for evolutionary optimization strategies, originally developed for the original domain, to be effectively transferred and integrated into frameworks operating on sparsified domains.

Table 2: Performance comparison (Mean and Standard Deviation (SD)) of different evolutionary optimization methods across various networks.

| Networks | Single-Domain | | | | | Multi-Domain | |
|---|---|---|---|---|---|---|---|
| | **SAEP** [22] | **CoeCo** [38] | **SSR** [39] | **Spars-D-EO** | **Spars-C-EO** | **Co-EO** | **Co-MLLM(KK)** |
| **USAir** | 41.1±2.89(-) | **48.9±0.50**(+) | 48.4±1.04(+) | 41.5±3.25(-) | 40.9±2.62(-) | 44.5±2.17(-) | 45.3±2.85 |
| **Netscience** | 14.9±0.79(-) | 16.8±0.24(-) | 16.4±0.26(-) | 14.1±0.78(-) | 14.2±0.65(-) | 15.3±0.46(-) | **17.6±0.70** |
| **Polblogs** | 189.8±6.99(-) | **223.5±2.56**(+) | 198.6±7.86(-) | 192.2±6.86(-) | 188.5±10.59(-) | 203.7±9.21(≈) | 207.4±4.05 |
| **Facebook** | 387.9±30.54(≈) | 378.8±3.28(-) | 383.7±26.98(-) | 326.5±14.38(-) | 337.2±30.93(-) | 367.2±25.19(-) | **396.3±31.90** |
| **WikiVote** | 492.2±13.16(-) | 502.0±8.36(-) | 504.2±12.17(-) | 489.1±23.63(-) | 478.6±25.04(-) | 518.3±17.76(≈) | **524.9±16.76** |
| **MSU24** | 1119.9±44.56(-) | 1115.2±21.83(-) | 1118.1±45.62(-) | 1111.6±46.19(-) | 1057.0±44.86(-) | 1133.1±28.38(≈) | **1146.7±18.53** |
| **Texas84** | 1992.3±180.30(-) | 2515.7±148.16(≈) | 2198.2±173.29(-) | 1998.4±204.98(-) | 1969.5±229.25(-) | 2459.0±109.70(-) | **2522.9±111.28** |
| **Rutgers89** | 704.6±31.18(-) | 742.0±27.04 (-) | 712.0±29.39(-) | 707.3±41.70(-) | 705.0±18.91(-) | 736.7±28.21(-) | **765.9±22.10** |
| **Avg ranking** | 4.38 | 2.63 | 3.38 | 5.75 | 6.63 | 3.25 | 1.38 |

The strength of the cooperative mechanism can also be observed in Figure 4, which shows the optimization process of Spars-D, Spars-C and the cooperative way. Across all networks, the cooperative approach consistently outperforms the other two methods with higher fitness values, suggesting that the cooperative optimization strategy effectively leverages information from both sparsified domains to enhance optimization performance.

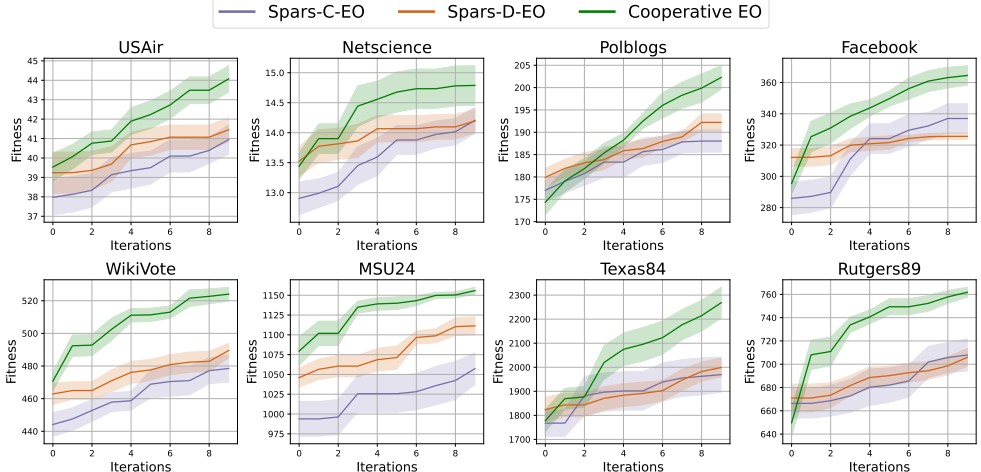

Figure 4: Comparison of cooperative and vanilla evolutionary optimization (EO). Shaded regions represent the Standard Error of the Mean (SEM). The number of solutions in each Co-EO subprocessor population is set to half that of the single-domain configuration, since two sparsified networks are employed.

Table 3: Comparison of cooperative optimization for different transfer intervals. The results are obtained based on the vanilla EO with 30 runs.

| Graph | Interval = 2 | Interval = 5 |
|-------|--------------|--------------|
| **USAir** | **44.4 ± 2.42** | 42.9 ± 2.24 |
| **Netscience** | **14.9 ± 0.77** | **14.9 ± 0.86** |
| **Polblogs** | **199.9 ± 7.88** | 192.4 ± 6.31 |
| **Facebook** | **368.5 ± 25.31** | 359.9 ± 26.94 |
| **WikiVote** | **527.5 ± 14.77** | 510.8 ± 15.37 |
| **MSU24** | **1154.6 ± 14.60** | 1140.2 ± 17.12 |
| **Texas84** | **2332.2 ± 179.56** | 2235.9 ± 174.71 |
| **Rutgers89** | **753.1 ± 17.67** | 743.2 ± 21.81 |

To further evaluate the effectiveness of our cooperative framework, we compare different configurations of the knowledge transfer interval. As shown in Table 3, more frequent knowledge transfer (Interval = 2) generally results in improved optimization performance, demonstrating the importance of timely information sharing: the elite solutions discovered in one sparsified domain can be effectively used to guide the search in other domains, and these cross-domain insights can jointly improve overall optimization outcomes.

## 4.2 Effectiveness Examination of Ensemble Approach

To show the strength of the ensemble method, we compare the ensemble variant of MLLM with its single-layout counterpart. Vanilla EO refers to the probability-based evolutionary optimization. Symbols indicate statistical significance from the Wilcoxon test (95% confidence interval): '+' for better, '≈' for no difference, and '−' for worse performance than MLLM-Ensemble. Table 4 provides a comparative analysis of different evolutionary optimization modes in optimizing influence maximization. As seen, MLLM-Ensemble achieves the highest influence scores across most networks. The ensemble approach benefits from combining multiple layout perspectives and employing a consensus voting strategy, making it more robust. In contrast, the other MLLM-based methods (KK, FR, and GraphOpt) show mixed performance, especially in Facebook and Texas84 networks, demonstrating the influence of layout on the performance of MLLM-based optimization.

Figure 5 presents the fitness progression of various MLLM-based optimization methods. Single-layout methods (KK, FR, and GraphOpt) exhibit network-dependent performance, for example, the FR layout performs significantly worse than the others in Polblogs and Facebook, while KK or GraphOpt underperform in different cases. These results highlight that no single layout consistently dominates across networks. By integrating multiple layouts, the ensemble method achieves steady improvement and avoids stagnation, thereby demonstrating superior robust optimization through diverse visual inputs. Additional evidence supporting the effectiveness of the ensemble strategy is presented in Figures 7 and 14, where two tasks are employed for further validation.

Table 4: Performance comparison (Mean and Standard Deviation (SD)) of different reproduction modes across various networks. The domain is built upon community-based sparsification method.

| Networks | Reproduction modes | | | | |
|---|---|---|---|---|---|
| | **Vanilla EO** | **MLLM-KK** | **MLLM-FR** | **MLLM-GraphOpt** | **MLLM-Ensemble** |
| **USAir** | 42.1±2.24(-) | 44.4±2.78($\approx$) | 44.9±2.70($\approx$) | 44.7±2.03($\approx$) | **45.1±2.10** |
| **Netscience** | 14.0±0.34($\approx$) | 14.6±0.67($\approx$) | **15.0±0.99($\approx$)** | 14.7±0.92($\approx$) | 14.9±0.63 |
| **Polblogs** | 190.0±6.31(-) | 191.8±6.16(-) | 187.3±6.23(-) | 191.0±7.22(-) | **196.3±6.92** |
| **Facebook** | 337.4±37.83(-) | 356.8±27.30(-) | 357.1±30.70(-) | 375.2±24.91($\approx$) | **387.3±31.15** |
| **WikiVote** | 466.2±27.28(-) | 483.3±24.08(-) | 474.7±13.75(-) | 475.2±22.54(-) | **510.0±10.30** |
| **MSU24** | 1053.2±46.85(-) | 1099.6±41.54(-) | 1117.6±16.40($\approx$) | 1126.7±23.39($\approx$) | **1129.2±14.04** |
| **Texas84** | 2075.0±297.49(-) | 2104.2±189.68(-) | 2269.2±171.70($\approx$) | 2205.8±229.82(-) | **2285.1±95.72** |
| **Rutgers89** | 705.6±36.87(-) | 749.6±11.27($\approx$) | 746.9±17.06($\approx$) | 742.9±20.42(-) | **757.9±7.99** |
| **Avg ranking** | 4.63 | 3.25 | 3.00 | 3.00 | **1.13** |

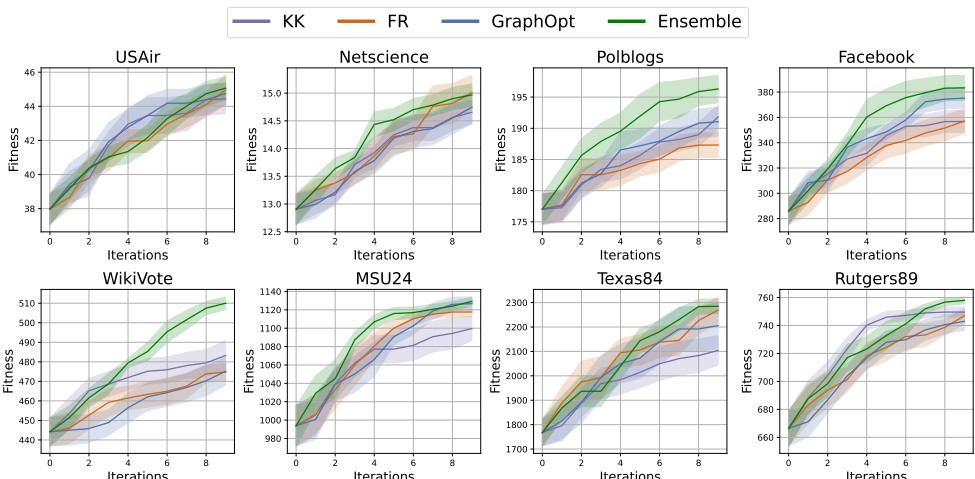

Figure 5: Comparison of MLLM-based optimization performance across different graph layouts. Shaded regions represent the Standard Error of the Mean (SEM).

## 4.3 Examination of Network Layout

Initialization can be viewed as a one-time optimization of the network to a certain extent, since the MLLMs are required to select the most promising candidate. By prompting the MLLM to initialize the population, we further investigate its structural sensitivity to network images generated under different layouts and sparsification strategies, as illustrated in Figure 6. Each point represents a single solution in the population, which is directly generated by the MLLMs as the initial population for optimization. Note that the sparsification strategy also influences the visualization, as the layout algorithm adaptively positions nodes based on structural properties.

As demonstrated in several cases, the choice of layout has a clear impact on MLLMs' performance. For instance, initial candidates generated with KK and GraphOpt exhibit higher overall fitness compared to those produced with FR in Polblogs and Facebook (Spars-C). Moreover, the influence of the sparsification strategy on MLLMs can also be observed. In networks such as USAir and Texas84, the same layout yields distinct distributions, indicating that the sparsification strategy is related to the layout.

## 4.4 Generalizability Analysis

To examine the generalizability of MLLM-based structure-aware optimization, we evaluate it on tasks with different characteristics. We first study a sequential decision-making problem in which nodes are removed to disrupt a network's connectivity [40], and its effectiveness is measured using the area under the curve (AUC) of the largest connected component (LCC). We compare the single-layout and multi-layout approaches on two non-sparsified networks (Dolphins and Lesmis) in Figure 7, and observe that the ensemble strategy consistently outperforms any single layout.

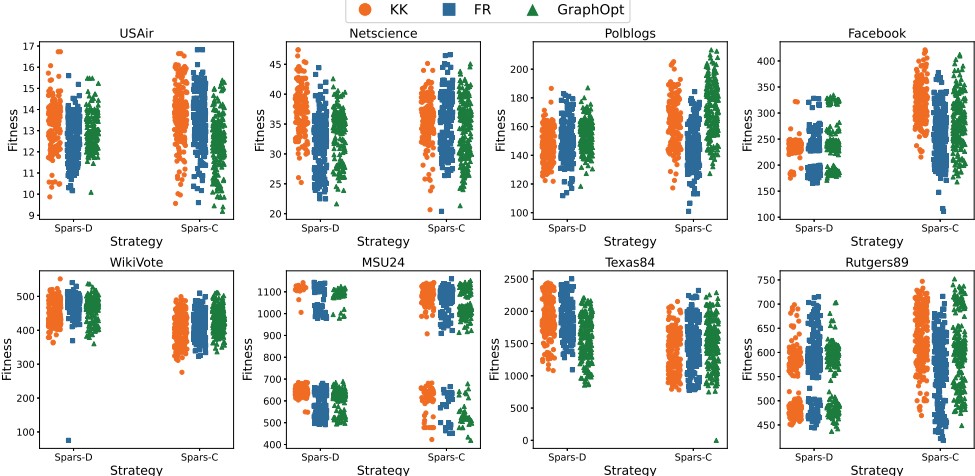

Figure 6: Distribution of MLLM-based initialization with different layouts and sparsified networks.

We then consider a classical permutation-based combinatorial problem, the Traveling Salesman Problem (TSP). In TSP, the layout is fixed by the coordinates of the cities, which makes the multi-layout ensemble inapplicable. Thus, we compare the vanilla evolutionary optimization with our MLLM-based evolutionary optimization, as shown in Figure 8. The results demonstrate that incorporating MLLM-guided evolutionary operators significantly improves search effectiveness. Together, these tasks, ranging from subset selection (one additional example can be found in Figure 14) in graphs to permutation optimization as well as sequential decision-making problem, illustrate the breadth of our framework, demonstrating comprehensive evidence of the generalizability and flexibility of the structure-aware optimization approach.

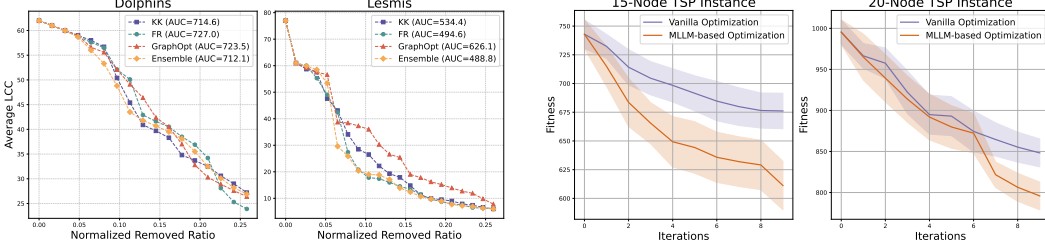

Figure 7: Comparison of different layout strategies in the network dismantling problem.

Figure 8: Comparison of different evolutionary optimization strategies in the TSP.

## 5 Conclusion

This paper investigates the methods to facilitate the integration between evolutionary optimization and multimodal large language models (MLLMs), aiming to harness their combined strengths. This synergy is significant as it opens new avenues for solving complex problems more effectively by leveraging the deep contextual understanding of MLLMs. By utilizing MLLMs as evolutionary operators, we demonstrate that incorporating graph sparsification and diverse network layouts significantly enhances the optimization process. Our findings also highlight the superiority of knowledge transfer across different sparsified networks in the cooperative evolutionary framework.

## Acknowledgement

This work was supported by the Ministry of Education (MOE) Singapore, under the Academic Research Fund (AcRF) Tier 1 Grant No. RS01/24. Kang Hao Cheong acknowledges additional support from the A*STAR Human Potential Programme Grant No. H23P1M0006.

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

# A  Discussion

## A.1  Representation Comparison of Natural Language and Image

In addition to the advantages of image-based encoding for graph-structured combinatorial optimization discussed in [10], we extend the discussion by presenting two additional aspects.

**Token usage:** In conventional LLM-based evolutionary optimization, the graph structure and candidate solution must be described separately in text. This inflates the token count, as both structural information and solution details have to be serialized into lengthy textual descriptions. As the problem size grows, the token cost quickly becomes prohibitive. By contrast, in image-based representation, the graph and solution are merged into a single visual object. This compact encoding substantially reduces token usage, since the MLLM processes the image as one unified input rather than parsing long text sequences.

**Reasoning:** When structural information and solution are separated into different textual descriptions, the LLM must internally re-align two distinct streams of data, i.e., the abstract topology of the network and the numerical value of the solution before performing inference. This disintegration increases reasoning difficulty and introduces unnecessary cognitive overhead. In image-based encoding, however, the network and solution are presented together in a single representation. This integration allows the model to reason about their interactions more directly and coherently, without the burden of reconstructing connections across fragmented inputs.

## A.2  Limitation and Mitigation

### A.2.1  Limitation

Although graph sparsification improves the scalability by reducing the size of large networks, it also has inherent constraints that limit its applicability.

**Subset selection problems:** In practice, plotting beyond roughly 200 nodes produces highly cluttered images, which in turn makes MLLM reasoning unreliable. This imposes a hard ceiling on the number of nodes that can be included in the visual representation. If the number of important or required selected nodes exceeds this threshold, the candidate node will also necessarily far exceed it, rendering the image-based approach invalid.

**Permutation problems:** In tasks such as permutation optimization, sparsification cannot be applied. These problems require complete output over all nodes in the original network, meaning no node can be safely removed. Any reduction of the graph would distort the problem definition, making the method inapplicable.

### A.2.2  Mitigation

To mitigate these limitations, we plan to explore divide-and-conquer strategies in the future work.

**Subset selection problems:** For subset selection tasks, one promising approach is to employ community detection to partition the large graph into smaller subgraphs. Each subgraph remains within the visualization threshold, allowing MLLMs to reason effectively over its structure and candidate solutions. The results from individual subgraphs can then be aggregated, ensuring that the final solution considers both local community-level optimization and global consistency.

**Permutation problems:** For permutation-type tasks, the graph can be partitioned into geographical or structural regions (e.g., clustering cities into local areas for TSP). The MLLM can then solve the permutation subproblem within each region before combining the partial tours into a global route. This regional divide-and-conquer strategy preserves all nodes while reducing the reasoning complexity faced by the model.

## A.3  Injection Strategy

Elite candidates identified in the master domain must be adapted before they can be injected into the subprocessors' local search spaces as the sparsified networks may differ in size and structure. After projection, some candidates may fall short of the required size $k$. To address this, we employ a

Table 5: Comparison of optimization performance for different injection strategies. The result is obtained based on the vanilla evolutionary optimization with 30 runs.

| Graph | Random filling | Heuristic-based filling |
|---|---|---|
| USAir | $44.2 \pm 2.30$ | $\mathbf{44.4 \pm 2.42}$ |
| Netscience | $14.8 \pm 0.86$ | $\mathbf{14.9 \pm 0.77}$ |
| Polblogs | $198.5 \pm 8.03$ | $\mathbf{199.9 \pm 7.88}$ |
| Facebook | $352.4 \pm 25.83$ | $\mathbf{368.5 \pm 25.31}$ |
| WikiVote | $521.2 \pm 17.75$ | $\mathbf{527.5 \pm 14.77}$ |
| MSU24 | $1146.8 \pm 25.21$ | $\mathbf{1154.6 \pm 14.60}$ |
| Texas84 | $2312.1 \pm 156.01$ | $\mathbf{2332.18 \pm 179.56}$ |
| Rutgers89 | $732.9 \pm 27.02$ | $\mathbf{753.12 \pm 17.67}$ |

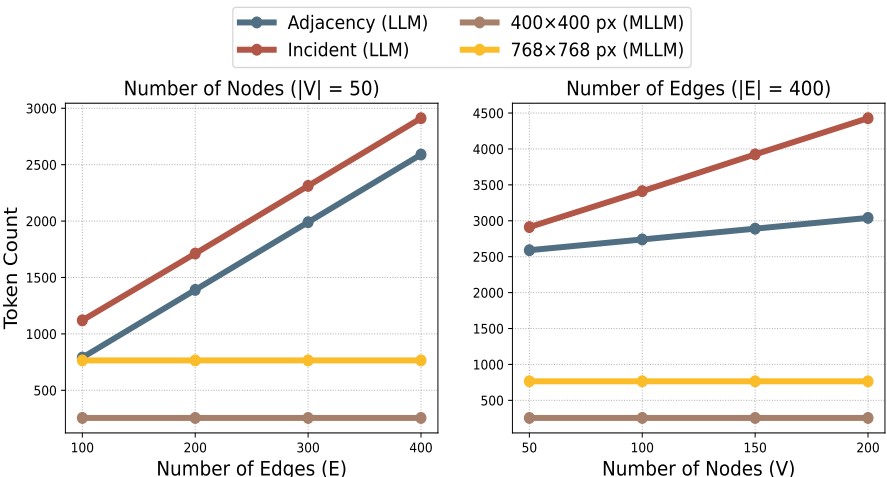

Figure 9: Comparison of token usage across different network depiction methods and settings [10].

filling step to ensure that each injected solution is feasible by supplementing it with additional nodes. Specifically, we explore two strategies for the filling step:

**Heuristic-based filling (default):** prioritizes nodes with high betweenness centrality.

**Random filling:** selects nodes randomly from all available nodes.

The results in Table 5 show that heuristic-based filling achieves slightly better performance than random filling across all tested graphs, though the differences are small. This suggests that while heuristics provide a marginal advantage, the cooperative framework itself is robust and consistently delivers good optimization results regardless of the injection strategy employed. Moreover, when compared to the single-domain results in Table 2, these findings further demonstrate the effectiveness of our cooperative framework.

# B    Computational Cost Analysis

While large language models (LLMs) have proven effective across a wide range of domains [41, 42, 43], they often struggle with discrete graph-structured problems [44, 45]. To address these challenges, researchers have explored the use of MLLMs, which leverage visual representations of graphs to enhance reasoning capabilities [46, 47]. For instance, recent work has demonstrated success in solving the traveling salesman problem by combining textual and visual modalities [48]. Building on this trend, our approach applies MLLM-based reasoning to the domain of discrete network optimization.

The scalability of the input representations is a critical factor when working with LLMs and MLLMs [10]. Traditional textual encodings, such as adjacency and incidence representations [49], incur a token cost that grows linearly with the size of the network, making them increasingly inefficient

as the number of nodes and edges expands. In contrast, visual inputs used in MLLMs maintain a constant token count, independent of the graph's scale (see Figure 9).

Table 6: Average running time (in seconds) per API call for different MLLMs performing crossover (with 2 input images) and mutation (with 1 input image) during evolutionary optimization.

| Networks | Kamada-Kawai | | Fruchterman-Reingold | | GraphOpt | |
|---|---|---|---|---|---|---|
| | Crossover | Mutation | Crossover | Mutation | Crossover | Mutation |
| gpt-4o-2024-11-20 | 3.5±0.66 | 2.4±0.74 | 3.4±0.68 | 2.3±0.73 | 3.6±0.81 | 2.3±0.49 |
| Gemeni-2.0-flash-lite | 2.2±0.44 | 2.2±0.49 | 2.3±0.51 | 2.1±0.53 | 2.3±0.45 | 2.2±0.52 |
| Qwen-vl-max | 3.8±1.15 | 2.3±1.25 | 3.7±1.58 | 2.3±1.36 | 3.8±1.15 | 2.3±1.17 |

We compare the running time across different models in performing crossover and mutation under three layouts in Table 6, which presents the average API call running time, revealing several key insights. First, the running time clearly depends on the foundation model, with Gemeni-2.0-flash-lite consistently being the fastest and Qwen-vl-max generally incurring higher latency. Second, consistent with findings from [10], crossover operations requiring two input images are more time-consuming than mutation operations, which involve only one image. This trend holds across all models and layout strategies. Third, despite using the same foundation models, the running times reported here differ from those in [10], potentially due to variability in server load, highlighting the importance of deploying these models in practical environments.

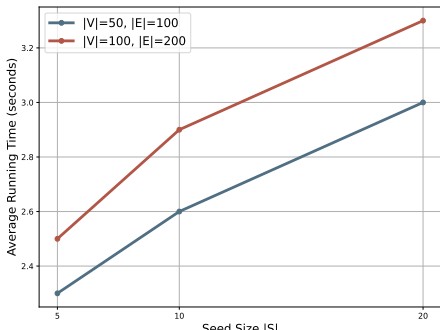

Figure 10: Average running time (in seconds) per API call for MLLMs performing initialization across different settings of network and seed size.

Figure 10 shows the average running time per API call for MLLMs (gpt-4o-2024-11-20) during initialization under different network and seed size settings. The results indicate that the running time is influenced by both context difficulty, represented by network size ($|V|$, $|E|$), and task difficulty, represented by solution size ($|S|$). Larger networks and required seed sizes consistently lead to longer running times, reflecting the increased computational demands of more complex inputs.

## C  Fidelity Validation of MLLMs Outputs

While MLLMs demonstrate promising capabilities in reasoning over visualized network structures, their outputs may still violate task-specific constraints. Therefore, we implement a set of fidelity checks at each evolutionary stage (initialization, crossover, and mutation) to detect and correct such issues. These checks are designed to maintain the structural integrity, constraint compliance, and overall quality of the evolving population. The validation criteria applied in each stage are detailed below.

The outputs of MLLMs are validated at different evolutionary stages to ensure solution quality and adherence to constraints:

**Initialization Phase:**

- $T_I^1$ - **Valid Node Check**: Ensures that all nodes belong to the predefined valid set.
- $T_I^2$ - **Initialization Size Check**: Verifies that the initial solutions meet the required seed size.

Table 7: Evaluation of MLLM-generated outputs at initialization under the Spars-D setting.

| Networks | Spars-D (KK) | | | Spars-D (FR) | | | Spars-D (GraphOpt) | | |
|---|---|---|---|---|---|---|---|---|---|
| | $T_I^1$ | $T_I^2$ | $T_I^3$ | $T_I^1$ | $T_I^2$ | $T_I^3$ | $T_I^1$ | $T_I^2$ | $T_I^3$ |
| **USAir** | 100% | 100% | 99.8% | 100% | 100% | 100% | 100% | 100% | 99.9% |
| **Netscience** | 100% | 100% | 96.8% | 100% | 100% | 94.8% | 100% | 100% | 99.9% |
| **Polblogs** | 100% | 100% | 99.7% | 100% | 100% | 98.9% | 100% | 100% | 99.6% |
| **Facebook** | 100% | 100% | 99.3% | 99.9% | 100% | 96.3% | 99.9% | 100% | 96.8% |
| **WikiVote** | 100% | 100% | 97.8% | 100% | 99.5% | 97.2% | 100% | 100% | 98.3% |
| **MSU24** | 100% | 100% | 97.7% | 100% | 100% | 91.6% | 99.9% | 100% | 99.3% |
| **Texas84** | 100% | 100% | 98.7% | 100% | 100% | 99.1% | 100% | 100% | 98.8% |
| **Rutgers89** | 100% | 100% | 98.1% | 100% | 100% | 96.3% | 100% | 100% | 96.8% |

Table 8: Evaluation of MLLM-generated outputs at initialization under the Spars-C setting.

| Networks | Spars-C (KK) | | | Spars-C (FR) | | | Spars-C (GraphOpt) | | |
|---|---|---|---|---|---|---|---|---|---|
| | $T_I^1$ | $T_I^2$ | $T_I^3$ | $T_I^1$ | $T_I^2$ | $T_I^3$ | $T_I^1$ | $T_I^2$ | $T_I^3$ |
| **USAir** | 100% | 100% | 99.2% | 100% | 100% | 99.1% | 100% | 100% | 99.0% |
| **Netscience** | 100% | 100% | 96.2% | 100% | 100% | 96.0% | 100% | 100% | 96.1% |
| **Polblogs** | 100% | 100% | 99.7% | 100% | 100% | 98.8% | 100% | 100% | 99.5% |
| **Facebook** | 100% | 100% | 93.0% | 99.9% | 100% | 89.7% | 100% | 100% | 90.2% |
| **WikiVote** | 100% | 100% | 98.0% | 100% | 100% | 97.5% | 100% | 100% | 98.7% |
| **MSU24** | 100% | 100% | 98.9% | 100% | 100% | 98.9% | 100% | 100% | 99.1% |
| **Texas84** | 100% | 100% | 98.3% | 100% | 100% | 96.8% | 100% | 99.5% | 97.5% |
| **Rutgers89** | 100% | 100% | 95.1% | 100% | 100% | 97.5% | 100% | 100% | 98.8% |

- $T_I^3$ - **Low Degree Node Check**: Identifies nodes with low degrees that may affect solution quality.

**Crossover Phase:**

- $T_C^1$ - **Crossover Size Check**: Ensures that the offspring solutions adhere to size constraints.
- $T_C^2$ - **Duplicate Node Check**: Detects repeated nodes within the solution.
- $T_C^3$ - **Parent Node Source Check**: Confirms that all nodes in the offspring are inherited from parent solutions.

**Mutation Phase:**

- $T_M^1$ - **Node Presence Check**: Ensures that nodes targeted for removal actually exist in the solution.
- $T_M^2$ - **Mutation Valid Node Check**: Verifies that newly added nodes are part of the valid node set.
- $T_M^3$ - **Mutation Repetitive Node Check**: Checks whether added nodes are already present in the solution.

Due to the concern about hallucination, we examine the correctness of MLLM outputs during initialization in Tables 7 and 8. The results indicate that $T_I^1$ and $T_I^2$ achieve 100% accuracy across all networks and strategies, confirming that the MLLM-based evolutionary process consistently selects nodes from the valid set and adheres to the required seed size constraints. This demonstrates that the initialization process is robust across different sparsification and layout strategies. However, $T_I^3$ (Low-Degree Node Check) exhibits slight variations depending on the sparsification and layout used. The degree-based sparsification approach generally achieves higher $T_I^3$ accuracy, with values close to or exceeding 99% across all networks. In contrast, community-based sparsification shows slightly lower $T_I^3$ scores, especially in layouts like FR and GraphOpt, where percentages sometimes drop below 90% (e.g., Facebook: 89.7% in FR, 90.2% in GraphOpt). The differences in $T_I^3$ scores across layouts further suggest that graph layout sensitivity should be carefully considered when designing optimization strategies for influence maximization.

Figure 11 illustrates the impact of graph layout choices on the validation rates of MLLM-generated solutions across various networks during evolutionary optimization. Overall, while most validation

tests maintain high fidelity, noticeable discrepancies arise during the mutation phase especially for test $T_M^3$ (Mutation Repetitive Node Check). Networks such as Facebook, Polblogs, WikiVote, and Texas84 show significant variation in this metric depending on the layout used, with darker shades indicating more frequent validation failures. Notably, layouts like GraphOpt and KK are more susceptible to producing invalid outputs in this stage, particularly for complex networks (e.g., Polblogs and Facebook). These results highlight the sensitivity of MLLMs to visual layout encodings, and further justify the use of layout ensembles to enhance robustness and consistency in MLLM-based network optimization.

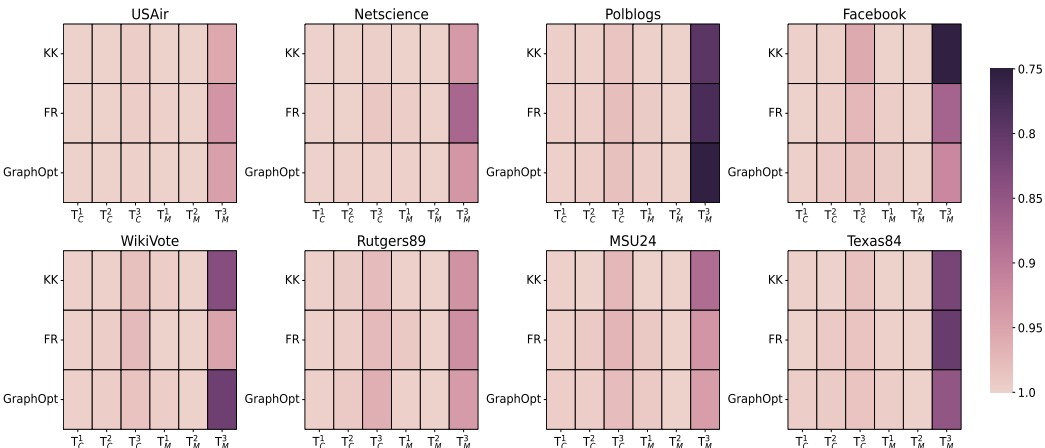

Figure 11: Examination of layouts' influence on validation rates for MLLMs across various networks. The sparsification strategy is the community-based graph sparsification. Higher validation rates indicate greater reliability of reproduction operations, with darker colors corresponding to lower validation rates.

Figure 12 highlights the impact of different sparsification strategies, Spars-C (community-based) and Spars-D (degree-based), on the validation rates of MLLM outputs across various networks. Consistent with findings in Figure 11, most validation checks yield high accuracy across strategies, except for $T_M^3$ (Mutation Repetitive Node Check), where darker shades indicate more frequent violations. Degree-based sparsification (Spars-D) tends to result in lower validation rates in this mutation phase, especially in networks like Netscience, WikiVote, Rutgers89, and Texas84 while Spars-C produces more errors in Polblogs and Facebook. This is because sparsification alters the structure and density of the graph, it inherently affects how layouts are rendered, potentially exacerbating the sensitivity of MLLMs to visual encoding. Thus, the interplay between sparsification and layout further underscores the importance of using ensemble and cooperative framework when applying MLLMs to discrete graph optimization tasks.

Moreover, we evaluate the structural sensitivity of MLLMs in network analysis by examining their mutation behavior. Ideally, mutations replace low-impact seeds with high-influence candidates. As shown in Figure 13, added nodes consistently exhibit higher degrees than removed ones across networks and layouts, indicating effective refinement and the excellent structural awareness of MLLMs.

# D Prompt Engineering

The prompts used in this work are presented in Tables 9–12. The prompt consists of two main components: (1) the *context-setting prompt*, which introduces the input information and specifies the roles of the agents, and (2) the *output-directive prompt*, which defines the desired output format and provides the necessary guidance or restrictions. Owing to the distinct characteristics of each problem, the prompts differ as follows:

**Influence Maximization:** The prompt is structured around three phases of evolutionary optimization: initialization, crossover, and mutation. The mutation phase is further divided into two steps: identifying the least influential node for removal and selecting the most promising node for addition.

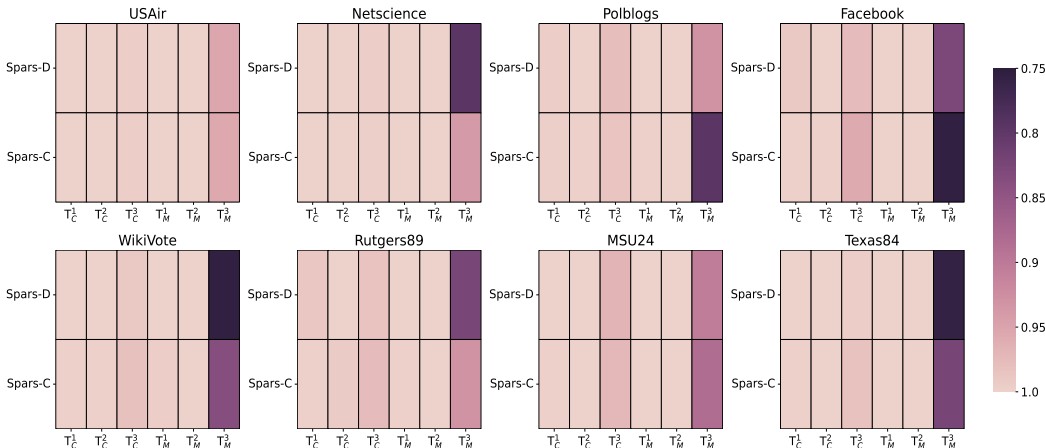

Figure 12: Examination of sparsification strategy's influence on validation rates for MLLMs across various networks. Higher validation rates indicate greater reliability of reproduction operations, with darker colors corresponding to lower validation rates.

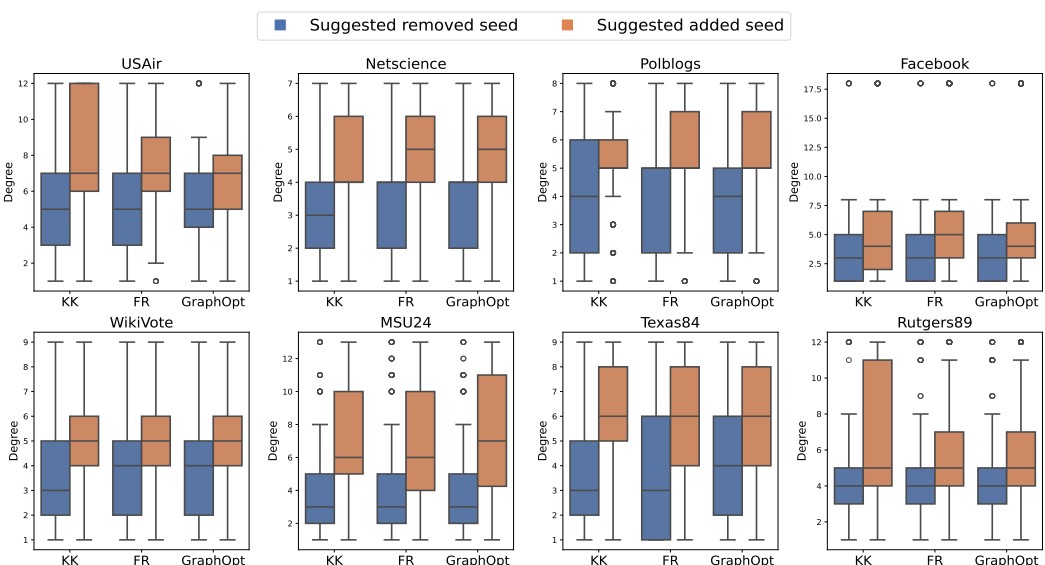

Figure 13: Comparison of degree distribution for removed and added seed nodes during MLLM-based mutation. The simplified graph is obtained by the community-based sparsification strategy.

**Immunization:** Given its similarity to influence maximization as a subset selection problem, the prompt design largely follows the same structure.

**Graph Dismantling:** As this is a sequential decision-making problem (used to validate the ensemble strategy), evolutionary optimization is not applied. Instead, the prompt directs the MLLM to identify the node most likely to cause network collapse.

**Traveling Salesman Problem (TSP):** Since this task requires outputting all nodes in the network, the prompt additionally provides the solution in textual form. This design reduces reasoning complexity and enhances solution feasibility.

Table 9: Structure of prompts for MLLM-based evolutionary operators of different phases.

| Task | Context-setting prompt | Output directive prompt |
|---|---|---|
| **Initialization** | You are an expert in network science and will be provided with one network in the form of an image. Please help me intelligently select nodes as the diffusion seeds in this network to achieve influence maximization. | Only provide a list of node indices separated by commas. |
| **Crossover** | Examine a network image where seed nodes are distinctly labeled. Carefully analyze the seed nodes present in each network image and suggest an optimal set of seed nodes that harness the advantages of both parent networks to maximize influence spread. | Focus on selecting high-degree nodes or nodes in strategic positions that significantly enhance network connectivity. Provide your answer as a list of node indices, separated by commas. |
| **Mutation (Removal)** | Examine a network image where seed nodes are colored and non-seed nodes are labeled in white. Identify the current seed node that contributes the least to influence maximization. | Focus on nodes that appear trivial or less connected. Provide the index of this node. |
| **Mutation (Addition)** | Examine a network image where seed nodes are colored and non-seed nodes are labeled in white. Propose a non-seed node that could significantly increase the network's influence spread. | Focus on nodes with higher degrees or strategically critical positions in the network. Provide the index of this node. |

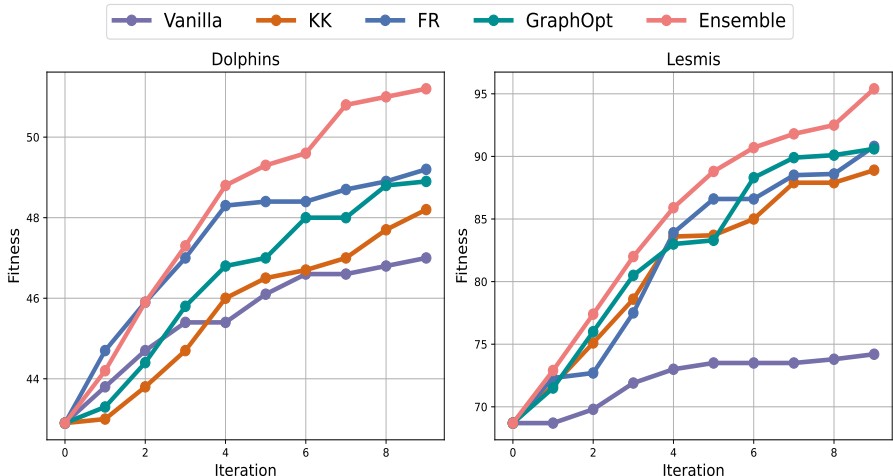

Figure 14: Comparison of different optimization methods in the network immunization problem.

Table 10: Structure of prompts for MLLM-based evolutionary operators of different phases for network immunization.

| Task | Context-setting prompt | Output directive prompt |
|---|---|---|
| **Crossover** | You are given two parent immunization strategies shown as graphs with highlighted nodes. Each highlighted node is selected for immunization. Generate a child solution by combining high-impact nodes from both parents. Prioritize nodes that connect to many non-immunized nodes, i.e., these create more cut edges, helping to stop virus spread. | Avoid clustering immunized nodes together. Your solution must include the same number of non-repetitive immunized nodes as the parents. Provide your answer as a list of node indices, separated by commas. |
| **Mutation (Removal)** | You are given an immunization solution visualized as a graph with highlighted nodes. To refine the solution, remove one node that contributes few connections to the rest of the network (i.e., nodes mostly surrounded by other immunized nodes or on the periphery). | The goal is to improve efficiency: keep only the most impactful nodes for maximizing the number of cut edges. Provide the index of this node. |
| **Mutation (Addition)** | You are given an immunization solution visualized as a graph with highlighted nodes. Improve the solution by adding one new node to immunize. | Focus on nodes that, when immunized, will connect to many still-vulnerable nodes and increase the total number of cut edges. Provide the index of this node. |

Table 11: Structure of prompts for MLLM-based network dismantling.

| Task | Context-setting prompt | Output directive prompt |
|---|---|---|
| **Network Dismantling** | You are an expert in network science and you will be provided with a network in the form of image. Each node is labeled with its node id in black text. Your task is to help me dismantle this network. | Please tell me which node to remove to most likely collapse this network, i.e., make the largest connected component as small as possible. |

# E   Visualization

The visualization of network is supported by the external library and the details of visualization are presented in Table 13. In this work, we adopt an ensemble approach by visualizing each network using three distinct layout algorithms before processing with MLLMs. This design choice is motivated by the observation that MLLMs are highly sensitive to spatial structure in visual inputs: Small variations in node positioning and edge arrangement can significantly influence the model's perception and output quality. To mitigate this layout-induced variability and enhance robustness, we incorporate multiple layouts during optimization. The layouts used in this study are as follows:

**Fruchterman-Reingold (FR)**: A force-directed algorithm where vertices repel each other and edges act like springs pulling connected nodes together. It tends to produce aesthetically pleasing layouts for medium-sized graphs.

**Kamada-Kawai (KK)**: Focuses on preserving the graph-theoretical distances between nodes. It minimizes an energy function to position nodes such that their Euclidean distances reflect their shortest path distances.

**GraphOpt**: A force-directed method that uses simulated annealing techniques to optimize node placement. It is tailored for larger networks and emphasizes computational efficiency while aiming for visually distinct clusters.

Table 12: Structure of prompts for MLLM-based evolutionary operators of different phases for Traveling Salesman Problem (TSP).

| Task | Context-setting prompt | Output directive prompt |
|---|---|---|
| **Crossover** | You are given two parent TSP tours, and each tour is shown as a image with numbered nodes. You are also given their visiting orders as lists of node IDs below.

Parent A tour: {solution}

Parent B tour: {solution} | Use these to create a new child tour that combines parts from both parents. Try to keep the path smooth and avoid long jumps. The child tour must visit every node exactly once and return to the start. Return the complete visiting sequence as an ordered list of node IDs. |
| **Mutation** | You are given a TSP tour: it is shown as a image with node numbers. You are also given its visiting order as a list of node IDs below.

Current tour: {solution} | Make a small change to the tour to try and shorten the overall path. You can swap two nodes, move one node to a different place, or reverse a small segment. The new tour must still visit every node once and return to the start. Return the complete visiting sequence as an ordered list of node IDs. |

Table 13: Visualization settings summary.

| Setting | Value |
|---|---|
| **Graph Library** | `plotly.graph_objs` |
| **Node Size** | 35 |
| **Label Size** | 22 |
| **Canvas Size** | $1200 \times 1200$ px |
| **Color** | Init. & Crossover: #2F7FC1
Mutation: Solution#2F7FC1,
Non-solution#FFFFFF |
| **Node Labels** | Init. & Mutation: All nodes
Crossover: Solution only |

