# OpenReview forum: "Structure-Aware Cooperative Ensemble Evolutionary Optimization on Combinatorial Problems with Multimodal Large Language Models"
_NeurIPS.cc/2025/Conference — NeurIPS 2025 poster_

### Official Review · Reviewer_MddL · 2025-07-01

**Clarity:** 3
**Significance:** 4
**Originality:** 3
**Rating:** 5
**Confidence:** 4

**Summary:**

This paper proposes a novel Structure-Aware Cooperative Ensemble Evolutionary Optimization framework that integrates Multimodal Large Language Models (MLLMs) into evolutionary optimization for solving combinatorial problems on complex networks. The key idea is to use image-based encodings of network structures, allowing MLLMs to serve as evolutionary operators (crossover and mutation), while incorporating graph sparsification , ensemble strategies, and cooperative frameworks to enhance robustness, reduce bias, and improve performance.

**Questions:**

(1)How generalizable is your framework to other combinatorial graph optimization tasks such as node cover, traveling salesman problem, or community detection?
(2)Table 5 shows significant differences in running time and performance across GPT-4o, Gemini, and Qwen. How does the choice of MLLM fundamentally affect the search behavior and final solution quality?
(3)The simplified networks used have only 50 nodes and 100 edges. How does the method scale to graphs with thousands or millions of nodes? What architectural changes would be required?
(4)How does the proposed MLLM-driven framework compare with traditional EA methods that use learned or hand-crafted mutation/crossover operators on graph data?

**Ethical Concerns:**

["NO or VERY MINOR ethics concerns only"]

**Final Justification:**

The reviewer has provided a rigorous proof in theory, and the work is based on solid foundations.

**Limitations:**

yes

**Paper Formatting Concerns:**

The writing of the paper is relatively standardized.

**Quality:**

3

**Strengths And Weaknesses:**

Main strengths
(1)	Novel Integration of MLLMs with Evolutionary Algorithms: The paper presents a creative and promising direction by using MLLMs not just as tools for reasoning or generation, but as core components in an evolutionary optimization pipeline.
(2)	Ensemble and Cooperative Framework Design: The proposed cooperative framework with multiple sparsified views and layout ensembles shows a strong understanding of how to mitigate MLLM limitations such as sensitivity to input presentation and structural bias.
(3)	Comprehensive Experimental Evaluation : The evaluation on eight real-world networks, along with ablation studies and statistical significance tests, provides solid empirical support for the claims made.

Main weakness
(1)	Limited Theoretical Contribution: While the framework is empirically sound, there is little theoretical analysis or formal guarantees about convergence, optimality, or generalization beyond the specific task of influence maximization.
(2)	Lack of Broader Applicability Discussion: The experiments focus solely on influence maximization; it’s unclear how easily the framework can be adapted to other combinatorial optimization tasks on graphs.
(3)	Computational Cost and Scalability Concerns: Although the paper discusses computational efficiency, the cost of repeated MLLM API calls and the scalability to much larger graphs are underexplored.

---

> ### Author Rebuttal · Authors · 2025-07-31
>
> ### **Response to # Reviewer Mddl**
>
> We sincerely thank you for the detailed evaluation and for highlighting the novelty, architectural soundness, and thorough empirical validation of our proposed framework. We have taken your comments seriously and addressed the theoretical limitations, scalability, and generalization concerns. Specifically, we have added results on new combinatorial tasks (e.g., TSP, network dismantling, network immunization), clarified the impact of MLLM selection, discussed architectural scalability strategies, as well as comparisons with hand-crafted evolutionary operators.
>
> ---
>
> ### **W1: Concern on limited theoretical analysis or formal guarantees**
>
> **Response:**
> Thank you for this important comment. Combinatorial optimization on graphs is inherently NP-hard, especially in the context of large-scale and non-linear networks. As such, no existing method including traditional or learning-based heuristics can provide general guarantees of optimality or convergence in these settings. Our primary focus, therefore, is on designing a practical and empirically effective framework that leverages the strengths of MLLMs for structure-aware search, and demonstrates consistent performance improvements across diverse problem domains. The experimental results in our work show that the proposed approach significantly enhances optimization quality compared to both traditional and baseline methods.
>
> We seek your kind understanding that this limitation is not specific to our framework, but reflects a broader challenge shared by all existing work operating on NP-hard problems due to inherent complexity of such kind of problem.
>
> Regarding generalization beyond influence maximization, we have conducted additional experiments on three different tasks (network immunization, TSP, and network dismantling), as detailed in our response to  **W2 & Q1**. These results further support the adaptability of our framework across a wide range of combinatorial problems.
>
> ---
>
> ### **W2 & Q1: Concern on unclear generalizability to other combinatorial graph tasks**
>
> **Response:**
> Thank you for your valuable comment. We fully recognize the importance of demonstrating the generalizability of our method. We have added more experiments to show the generalizability of our method as below:
>
> - **Network Immunization**: A critical problem in network science involving node removal to minimize epidemic spread [5].
> - **Traveling Salesman Problem (TSP)**: A classical permutation-based combinatorial problem.
> - **Network Dismantling**: A sequential decision-making problem where nodes are removed to break down a network's connectivity [6].
>
> Due to word limitations, the results for the three additional tasks have been provided in our response to **#Reviewer 11CR**. We kindly refer you there for full details.
>
> These tasks differ in structure and computational nature, ranging from subset selection to permutation optimization to sequential strategies, thus providing a comprehensive validation of our structure-aware optimization framework’s generality. Moreover, our ensemble layout strategy has also been evaluated in both the network immunization and dismantling tasks, where it consistently demonstrated superior robustness and solution quality compared to single-layout baselines.
>
> Collectively, these new experiments reinforce that our proposed MLLM-based evolutionary optimization framework is broadly applicable across a wide spectrum of combinatorial graph problems, not limited to influence maximization.
>
> ---
>
> ### **W3: Concern on computational cost and scalability of repeated MLLM calls**
>
> **Response:**
> Thank you for raising this concern. To date, there are already many interests using LLM to do optimization. As shown in **Figure 7**, scalability is one of advantages of our methods in term of API call cost.
>
> First, compared to traditional token-based LLM encodings, where input size grows linearly (or worse) with the number of nodes and edges, our visual encoding approach decouples input size from graph scale. Regardless of whether the graph contains 1,000 or 100,000 nodes, the number of tokens used during inference is determined solely by the size of the rendered image, which remains fixed. **This leads to a constant token cost per inference, independent of the graph’s scale.**
>
> To ensure effective and interpretable visualizations for larger graphs, our framework incorporates a **graph sparsification** step that selectively retains the most structurally important components. This approach preserves the essential features needed for optimization while keeping the visual input compact and suitable for MLLM processing. As a result, **our method achieves both efficient cost and scalability, even when applied to large graphs.**
>
> ---
>
> ### **Q2: Question on impact of MLLM choice on search behavior and performance**
>
> **Response:**
> Thank you for this insightful question. The choice of multimodal large language model (MLLM) indeed affects both the runtime and the quality of optimization outcomes. As shown in Table 5 in the original manuscript, we observed differences across GPT-4o, Gemini, and Qwen in terms of response latency and solution effectiveness.
>
> - **Runtime**: Variability mainly stems from differences in external factors such as API server load, rate limits, or backend infrastructure. These are more about system-level variations.
> - **Search behavior**: Each MLLM differs in its visual-textual reasoning ability, prompt understanding, and response precision. These differences fundamentally influence the evolutionary search trajectory, as MLLMs are responsible for mutation and crossover decisions. A more capable model can better interpret graph structure, leading to more effective candidate generation and faster convergence.
>
> ---
>
> ### **Q3: Question on scalability from simplified to large-scale networks**
>
> **Response:**
> Thank you for this question. While the simplified networks we visualize may contain as few as 50 nodes and 100 edges, they are not arbitrary samples, but rather structurally representative miniatures of the original networks. This design choice is grounded in the scale-free property of real-world networks [1], where a small subset of high-degree hub nodes governs much of the network’s dynamics, and the majority of nodes contribute marginally. This allows us to sparsify the network while preserving its functional backbone, which is a key contribution of our framework.
>
> Indeed, regardless of the original graph size, whether thousands or millions of nodes, our pipeline can scale by selectively retaining the structurally important components through graph sparsification, keeping the input size manageable for MLLMs while retaining optimization effectiveness.
>
> In our experiments, we work with networks containing up to **36,000+ nodes**, which is considered large-scale in the context of evolutionary optimization on graphs, and particularly significant for LLM/MLLM-based methods, where computational constraints are more pronounced.
>
> If scaled further to networks with millions of nodes, the same architectural design
> `Graph Sparsification → Visualization → MLLM-based Optimization`
> remains valid. For additional scalability, our framework can be extended with divide-and-conquer techniques, where the network is partitioned into overlapping subgraphs that are optimized independently and later integrated.
>
> ---
>
> ### **Q4: Concern on comparison with traditional or learned EA operators**
>
> **Response:**
> Thank you for this insightful question. We have compared our MLLM-driven framework with several state-of-the-art hand-crafted evolutionary optimization methods specifically designed for graph data. The results of this comparison are presented in Table 1 of the main paper and include the following baselines:
>
> - **SAEP**: IEEE TSMC (2023)
> - **CoeCo**: IEEE TEVC (2023)
> - **SSR**: IEEE TCSS (2022)
>
> Across multiple benchmark networks, our MLLM-based optimization consistently outperforms these hand-crafted methods in terms of solution quality. This result demonstrates the potential of leveraging multimodal large language models as a powerful and generalizable alternative to manually engineered or domain-specific evolutionary operators.
>
> Critically, our current implementation uses a simple and generic EO pipeline, without incorporating advanced mechanisms such as self-adaptive parameter tuning, multi-task optimization, or co-evolution strategies. Thus, our framework is extremely novel in its own rights and fully compatible with these enhancements. We expect that integrating them would further improve performance.
>
> To ensure methodological consistency across different combinatorial tasks, we carefully re-implemented our approach using a unified configuration for all experiments and have updated the results in  **Table 2** (and  **Figure 4**) of the original submission accordingly, i.e.,
>
> | Graph      | MLLM-Ensemble       |
> |------------|---------------------|
> | **USAir**      | 45.07 ± 2.10        |
> | **Netscience** | 14.97 ± 0.63        |
> | **Polblogs**   | 196.29 ± 6.92       |
> | **Facebook**   | 383.37 ± 31.15      |
> | **WikiVote**   | 509.99 ± 10.30      |
> | **MSU24**      | 1129.24 ± 14.04     |
> | **Texas84**    | 2285.11 ± 95.72     |
> | **Rutgers89**  | 757.94 ± 7.99       |
>
> The revised results, obtained under this consistent framework, continue to support our main claims regarding the method’s advantages. We will reflect these updates in the final version accordingly for completeness and reproducibility.
>
> ---
>
> We hope these updates and clarifications address your concerns. Please let us know if any further clarification is needed. We would be grateful if you would consider adjusting your score in light of the extended experiments and improvements.
>
> ---
>
> ### **Reference**
>
> [1] "Emergence of scaling in random networks." *Science* (1999).

---

> > ### Author Response · Authors · 2025-08-06
> >
> > Dear Reviewer Mddl,
> >
> > We would like to sincerely thank you again for recognizing the novelty of our work as your acknowledgment means a great deal to us.
> >
> > We understand that you may have other ongoing reviews or commitments, and we truly appreciate your time and attention.
> >
> > In our previous rebuttal, due to word limit constraints, we used cross-referencing to present our experimental results across **three different tasks** in response to Reviewer 11CR’s concerns regarding **generalizability**. We are glad to note that **Reviewer 11CR has acknowledged our efforts.**
> >
> > To make your review easier, we have now compiled and presented the experimental results directly in this thread. Please find them below:
> >
> > ### **Network Immunization**
> > A critical problem in network science involving node removal to minimize epidemic spread. We compared the optimization process across different evolutionary methods with Dolphins and Lesmis (The data can be found in “Network data” collected by Mark Newman). The optimization goal is to maximize the number of immune edges. The results further test our framework’s ability to handle structural node selection problems with global impact.
> >
> > **Dolphins:**
> >
> > | Dolphins         | 1st           | 3rd           | 5th           | 7th           | 10th          |
> > |------------------|---------------|---------------|---------------|---------------|---------------|
> > | **Normal**        | **42.9 ± 2.8** | 44.7 ± 2.6    | 45.4 ± 2.8    | 46.6 ± 1.8    | 47.0 ± 1.7    |
> > | **MLLM-FR**       | **42.9 ± 2.8** | 45.9 ± 3.6    | 48.3 ± 2.8    | 48.4 ± 2.8    | 49.2 ± 2.5    |
> > | **MLLM-KK**       | **42.9 ± 2.8** | 43.8 ± 2.2    | 46.0 ± 3.7    | 46.7 ± 3.7    | 48.2 ± 3.2    |
> > | **MLLM-GraphOpt** | **42.9 ± 2.8** | 44.4 ± 2.7    | 46.8 ± 4.5    | 48.0 ± 4.0    | 48.9 ± 4.2    |
> > | **MLLM-Ensemble** | **42.9 ± 2.8** | **45.9 ± 3.1** | **48.8 ± 4.1** | **49.6 ± 4.2** | **51.2 ± 3.9** |
> >
> > **Lesmis:**
> >
> > | Lesmis           | 1st           | 3rd           | 5th           | 7th           | 10th          |
> > |------------------|---------------|---------------|---------------|---------------|---------------|
> > | **Normal**        | **68.7 ± 7.9**    | 69.8 ± 7.3    | 73.0 ± 9.6    | 73.5 ± 9.3    | 74.2 ± 8.6    |
> > | **MLLM-FR**       | **68.7 ± 7.9**    | 72.7 ± 7.2    | 83.9 ± 7.0    | 86.6 ± 6.9    | 90.8 ± 9.6    |
> > | **MLLM-KK**       | **68.7 ± 7.9**    | 75.1 ± 10.5   | 83.6 ± 6.7    | 85.0 ± 7.4    | 88.9 ± 8.4    |
> > | **MLLM**          | **68.7 ± 7.9**    | 76.0 ± 7.8    | 83.0 ± 7.5    | 88.3 ± 9.7    | 90.6 ± 7.2    |
> > | **MLLM-Ensemble** | **68.7 ± 7.9**    | **77.4 ± 13.8**   | **85.9 ± 13.2**   | **90.7 ± 9.3**    | **95.4 ± 8.0**    |
> >
> > ---
> >
> > ### **Traveling Salesman Problem (TSP)**
> > A classical permutation-based combinatorial problem. Note that in TSP, the layout is fixed by coordinates, rendering the multi-layout ensemble inapplicable. Therefore, we compare the optimization process of normal evolutionary optimization and MLLM-based structure-aware evolutionary optimization to demonstrate that MLLM-guided evolutionary operators can significantly improve search effectiveness. The optimization goal is to minimize the route distance.
> >
> > | Instance 15 | 1st           | 3rd           | 5th           | 7th           | 10th          |
> > |-------------|---------------|---------------|---------------|---------------|---------------|
> > | **Normal**  | **742.90 ± 39.40** | 714.20 ± 48.60 | 698.40 ± 45.80 | 684.70 ± 54.10 | 676.10 ± 48.90 |
> > | **MLLM**    | **742.90 ± 39.40** | **683.60 ± 66.20** | **649.30 ± 69.40** | **635.80 ± 69.00** | **611.20 ± 66.80** |
> >
> > | Instance 20 | 1st            | 3rd            | 5th            | 7th            | 10th           |
> > |-------------|----------------|----------------|----------------|----------------|----------------|
> > | **Normal**  | **995.42 ± 46.93** | 957.56 ± 58.59 | 894.91 ± 75.14 | 874.44 ± 78.86 | 848.25 ± 53.82 |
> > | **MLLM**    | **995.42 ± 46.93** | **939.34 ± 90.82** | **891.93 ± 87.98** | **872.28 ± 75.98** | **795.83 ± 53.34** |
> >
> > ---
> >
> > ### **Network Dismantling**
> > A sequential decision-making problem where nodes are removed to break down a network's connectivity. Here we use this problem to examine the influence of single-layout and multi-layout to demonstrate the advantage of our ensemble framework. The presented results show the AUC of the largest network component, with the removal process terminating after 25% of the nodes were removed.
> >
> > | Model         | FR     | KK     | GraphOpt | Ensemble |
> > |---------------|--------|--------|----------|----------|
> > | **Dolphins**  | 726.95 | 714.60 | 723.50   | **712.15**   |
> > | **Lesmis**    | 494.60 | 534.40 | 626.10   | **488.80**   |
> >
> > We sincerely hope to engage with you further and would greatly appreciate your feedback.
> >
> > Warm regards,
> >
> > The Authors of Submission 25561

---

> > > ### Comment · Reviewer_MddL · 2025-08-06
> > >
> > > Thank you to the author for the response. My concerns have been resolved, and this is an excellent paper. Therefore, I will increase my final score.

---

> > > > ### Author Response · Authors · 2025-08-06
> > > >
> > > > Dear Reviewer MddL,
> > > >
> > > > Thank you very much for your kind words and for taking the time to review our work. We truly appreciate your thoughtful and constructive feedback throughout the process.
> > > >
> > > > We are especially honored by your recognition of our paper as excellent, and we sincerely thank you for your updated evaluation.
> > > >
> > > > Best regards,
> > > >
> > > > The Authors of Submission 25561

---

### Official Review · Reviewer_Etgw · 2025-07-05

**Clarity:** 3
**Significance:** 3
**Originality:** 4
**Rating:** 4
**Confidence:** 3

**Summary:**

Traditional encoding schemes challenge in capturing the intricate structural properties of networks. This paper propose using MLLMs as evolutionary operators to facilitate structure-aware optimization over graph data. The author demonstrates how MLLMs can function as evolutionary optimization operator and  explore the potential of this integration and how this integration can be further extended.

Experiments show that MLLM integration significantly outperforms single-layout approaches, and the collaborative mechanism further improves performance, achieving the best EDV scores on 8 real networks. The overall approach replaces traditional heuristics with visual large models, combining subgraph parallelization with knowledge transfer to balance accuracy and scalability.

**Questions:**

1. Your cooperative framework introduces several additional hyperparameters. Could you provide a sensitivity analysis or justify the choices of the default values?
2. While the method is claimed to be applicable to full-scale graphs, the experiments are conducted only on subgraphs. Could you include or comment on results using the complete network to support the scalability claim?
3. ANOVA is applied only during the initialisation stage. Have you considered performing a quantitative analysis of layout sensitivity for the crossover and mutation phases as well?
4. How does the overall optimisation time of your method compare to a traditional evolutionary optimization (EO) baseline without MLLM? Could you provide a runtime comparison?

**Ethical Concerns:**

["NO or VERY MINOR ethics concerns only"]

**Final Justification:**

The authors have already addressed most of my concerns. I would like to suggest an acceptance of this paper.

**Limitations:**

See weaknesses and questions above.

**Paper Formatting Concerns:**

no formatting concerns.

**Quality:**

3

**Strengths And Weaknesses:**

Strengths:
1. Pioneers feeding multi-layout graph visualisations into a multimodal LLM, using the model itself as the generator of evolutionary operators.

2. The end-to-end architecture, from sparsified multi-view subgraphs to cross-domain knowledge transfer, is conceptually sound and tightly aligned with the authors’ stated motivation.

3. Benchmarked on eight large real-world networks against diverse baselines, with rigorous statistical tests that convincingly show the added value of both the ensemble and cooperative components.

Weaknesses:

1. The cooperative framework adds several hyper-parameters, yet offers no sensitivity study or justification of the chosen defaults.

2. Although the method is claimed to work on full-scale graphs, all experiments use only subgraphs; at least one evaluation on the complete network is needed.

3. ANOVA is performed solely for the initialisation phase, with no quantitative layout-sensitivity analysis for crossover and mutation.

4. The overall optimisation time is not compared against a traditional EO baseline without MLLM.

---

> ### Author Rebuttal · Authors · 2025-07-31
>
> ### **Response to #Reviewer Etgw**
>
> We sincerely thank you for the comprehensive assessment. We are especially grateful for your recognition of the novelty, motivation alignment, and empirical rigor of our proposed framework. In response to your suggestions, we have conducted a sensitivity analysis on key hyperparameters, added a layout-sensitivity study for the reproduction phase, discussed the running time, and clarified the scalability of our framework.
>
> ---
>
> ### **W1/Q1: Concern on lack of hyperparameter sensitivity analysis in cooperative framework**
>
> **Response:**
> Thank you for raising this important point and we appreciate the suggestion to include parameter sensitivity to further improve our work. Among these parameters, the knowledge transfer frequency, i.e., how often elite solutions are exchanged across domains, is the most critical. In our default setting, this parameter is set to every 2 generations, which reflects a balance between sufficient cross-domain communication and preserving local evolutionary stability.
>
> In response, we have conducted a new sensitivity experiment by increasing the transfer interval to every 4 generations. The comparison results are provided in the table below. Our findings indicate that more frequent knowledge transfer (Interval = 2) leads to better optimization performance in most cases, supporting the effectiveness of our proposed cooperative optimization framework.
>
> | Graph      | Interval = 2       | Interval = 4       |
> |------------|--------------------|--------------------|
> | **Netscience** | **15.3 ± 0.46**    | **15.3 ± 0.47**    |
> | **USAir**      | 44.5 ± 0.17        | **44.7 ± 2.44**    |
> | **Polblogs**   | **203.7 ± 9.21**   | 193.2 ± 4.50       |
> | **Facebook**   | **385.6 ± 24.53**  | 376.3 ± 25.93      |
> | **WikiVote**   | **518.3 ± 17.76**  | 498.7 ± 19.28      |
> | **Rutgers89**  | **736.7 ± 28.21**  | 730.8 ± 16.87      |
> | **MSU24**      | **1133.1 ± 28.38** | 1108.1 ± 27.81     |
> | **Texas84**    | **2459.0 ± 109.70**| 2365.3 ± 184.24    |
>
> ---
>
> ### **W2 & Q2: Concern on evaluation using subgraphs instead of full-scale networks**
>
> **Response:**
> Thank you for this insightful comment. We would like to clarify that our method does not work on subgraphs (an independent part of the original network). Instead, we apply graph sparsification to the full network, preserving its core structural skeleton, particularly high-impact hub nodes based on the well-researched scale-free property of real-world networks. Importantly, this property ensures that even a reduced representation can retain most of the essential structural and functional characteristics of the original network.
>
> Our sparsified networks are thus not disconnected subgraphs, but rather purposefully constructed simplifications that retain high-fidelity approximations of influence, connectivity, and topology, enabling effective optimization while remaining computationally tractable for MLLMs. This design is fundamental to addressing the scalability challenge posed by large networks when visualized in their entirety. Rendering an entire network with tens of thousands of nodes on a single canvas results in severe visual clutter and degraded MLLM performance, something we explicitly seek to avoid.
>
> To support our scalability claim, we include a comparative experiment where candidate solutions are randomly initialized on the full, unsparsified network, and the sparsified network (as used in our framework). Using the same optimization framework in both cases, we observe that the sparsified version consistently yields better initial solution quality, reinforcing that sparsification not only improves scalability but also enhances optimization performance.
>
> | Graph      | Original          | Sparsified        |
> |------------|-------------------|--------------------|
> | **Netscience** | **16.24 ± 0.56**  | 15.89 ± 0.80       |
> | **USAir**      | 48.84 ± 2.23      | **49.27 ± 1.57**   |
> | **Polblogs**   | 183.32 ± 13.11    | **213.77 ± 5.94**  |
> | **Facebook**   | 236.46 ± 49.69    | **407.45 ± 8.86**  |
> | **WikiVote**   | 315.58 ± 28.19    | **545.13 ± 15.67** |
> | **Rutgers89**  | 327.80 ± 72.36    | **778.96 ± 25.65** |
> | **MSU24**      | 363.87 ± 102.34   | **1152.17 ± 27.81**|
> | **Texas84**    | 543.29 ± 147.81   | **2594.51 ± 144.18**|
>
> Furthermore, our utilised networks are up to **36,000+ nodes**, which are larger than those used in existing LLM/MLLM-based optimization works typically operating on small to medium-sized graphs.
>
> Our method’s scalability stems from its design: a modular pipeline of
> `Graph Sparsification → Visualization → MLLM-based Optimization`,
> which we believe is a practical and effective approach for handling large real-world graphs.
>
> ---
>
> ### **W3 & Q3: Concern on missing layout-sensitivity analysis for crossover and mutation**
>
> **Response:**
> Thank you for raising this important point. We acknowledge that our original layout-sensitivity analysis focused on the initialization phase, which is a one-time operation and more straightforward to evaluate statistically.
>
> In contrast, crossover and mutation are iterative operations that occur across multiple generations, and their effects are inherently accumulative and interdependent. This makes it challenging to isolate and assess the sensitivity of each reproduction step to layout choice using standard statistical methods. For example, the fitness outcome of a solution after the 1st generation may differ substantially from that after the 10th generation even under the same layout due to the influence of prior evolutionary history. As a result, comparing layouts at the same generation across independent runs does not guarantee a fair or controlled assessment.
>
> As a solution, we have conducted a statistical comparison between single-layout and ensemble approaches across the full optimization process, as shown in **Table 2**. These results demonstrate that layout differences significantly impact final optimization performance.
>
> ---
>
> ### **W4 & Q4: Concern on missing runtime comparison with traditional EO**
>
> **Response:**
> Thank you for raising this important question regarding runtime overhead. As shown in  **Table 5** of the appendix, the runtime per MLLM-based operation (crossover and mutation) depends on factors such as the model architecture (e.g., GPT-4o) and server load at inference time. These operations are inherently slower than traditional, code-based EOs, which can be executed in milliseconds due to their lightweight nature.
>
> However, it is important to note that traditional EO approaches are not structure-aware unless supplemented with manual, task-specific heuristics or engineered operators. In contrast, our MLLM-based method leverages multimodal perception to guide the search more intelligently, resulting in significantly improved optimization performance, as shown across multiple tasks and datasets.
>
> While we acknowledge the higher computational cost, we seek your kind understanding that this is not just a limitation of our framework, but a general challenge shared by all LLM-based optimization approaches. As multimodal models continue to improve in efficiency, we expect this gap to diminish over time.
>
> ---
>
> We hope these additions and clarifications have addressed your concerns. Please let us know if any further clarification is needed. We kindly request you to consider a score adjustment based on these improvements.

---

> ### Comment · Reviewer_Etgw · 2025-08-06
>
> Thank you for your response. I’m especially pleased to see that additional experiments have been included to address the first two concerns regarding hyperparameter sensitivity analysis and scalability. I also appreciate the clarifications provided for the last two concerns. Please incorporate the response into your manuscript.

---

> > ### Author Response · Authors · 2025-08-06
> >
> > Dear Reviewer Etgw,
> >
> > Thank you very much for your thoughtful response and for acknowledging our additional experiments and clarifications. We truly appreciate your recognition of our efforts to address the concerns regarding hyperparameter sensitivity and scalability.
> >
> > We will make sure to incorporate the relevant parts of our response into the revised manuscript, as per your suggestion.
> >
> > Thank you again for your valuable feedback and time.
> >
> > Best regards,
> >
> > The Authors of Submission 25561

---

### Official Review · Reviewer_11CR · 2025-07-18

**Clarity:** 3
**Significance:** 3
**Originality:** 3
**Rating:** 5
**Confidence:** 5

**Summary:**

This paper proposes an ensemble framework for structure-aware optimization on graph data, where multimodal large language models (MLLMs) are employed as evolutionary operators. The authors further incorporate the proposed framework with the following techniques for improvements:

-  Applying graph sparsification techniques to prune complex networks while preserving essential structural properties.
-  Utilizing a master-slave framework to distribute networks across multiple sparsified views, mitigating the biases produced by only viewing a single sparsified graph.

The experimental results on real-world networks validate the effectiveness of the proposed framework.

**Questions:**

- The paper applies a consensus voting mechanism among MLLMs. Have the authors considered a weighted voting variant, where the weights are proportional to the fitness scores of the best solutions produced by each MLLM?

- While the ensemble approach with image-based MLLMs helps mitigate layout sensitivity, it would be interesting to explore whether a single graph-based MLLM could achieve similar robustness without requiring multiple layout representations. Have the authors considered this alternative in the study?

- See Weaknesses for further comparisons on different combinatorial graph optimization tasks and SOTA methods with different modalities.

**Ethical Concerns:**

["NO or VERY MINOR ethics concerns only"]

**Final Justification:**

Increase the rating score to 5.
- The additional experiments addressing the generalizability concerns raised in W1/Q3 are clear and well-executed.
- The authors have provided a clear clarification regarding their future work in response to Q1.

**Limitations:**

Yes.

**Paper Formatting Concerns:**

No issues.

**Quality:**

3

**Strengths And Weaknesses:**

Strengths:
- The paper proposes an innovative algorithm that leverages MLLMs as evolutionary operators in the optimization process on graph data.
- The framework, which utilizes multiple sparsification views and various layouts to improve solution quality, is theoretically sound and effective, as evidenced by its better performance in ablation studies.
- The paper is well-organized, and the formulations of the entire framework are comprehensive and easy to follow.

Weaknesses:
- While the method is designed for combinatorial graph optimization, the experiments focus exclusively on the influence maximization task, which limits the insights into its generalizability. Including additional combinatorial optimization tasks in the experimental analysis would strengthen the evidence for its generalizability.
- The paper lacks comparisons with other state-of-the-art methods involving different modalities, such as the GNN-based methods GLIE [1] and AutoGNP [2], which are designed for combinatorial graph optimization.

References:

[1] Maximizing Influence with Graph Neural Networks

[2] Combinatorial Optimization with Automated Graph Neural Networks

---

> ### Author Rebuttal · Authors · 2025-07-31
>
> ### **Response to # Reviewer 11CR**
>
> We sincerely thank you for the encouraging comments and for recognizing the novelty and clarity of our framework. In response, we have conducted additional experiments on new varied combinatorial tasks (e.g., TSP, network immunization, dismantling) and addressed your suggestions regarding comparison with other methods.
>
> ---
>
> ### **W1/Q3: Concern on lack of diverse combinatorial task evaluation**
>
> **Response:**
> We highly appreciate your suggestion to demonstrate the broader applicability of our framework beyond influence maximization. Please be assured that we have now taken concrete steps to address it by conducting additional experiments on three diverse combinatorial optimization tasks.
>
> ---
>
> **Network Immunization**
> A critical problem in network science involving node removal to minimize epidemic spread [5]. We compared the optimization process across different evolutionary methods with Dolphins and Lesmis (The data can be found in “Network data” collected by Mark Newman). The optimization goal is to maximize the number of immune edges. The results further test our framework’s ability to handle structural node selection problems.
>
> **Dolphins:**
>
> | Dolphins         | 1st           | 3rd           | 5th           | 7th           | 10th          |
> |------------------|---------------|---------------|---------------|---------------|---------------|
> | **Normal**        | **42.9 ± 2.8** | 44.7 ± 2.6    | 45.4 ± 2.8    | 46.6 ± 1.8    | 47.0 ± 1.7    |
> | **MLLM-FR**       | **42.9 ± 2.8** | 45.9 ± 3.6    | 48.3 ± 2.8    | 48.4 ± 2.8    | 49.2 ± 2.5    |
> | **MLLM-KK**       | **42.9 ± 2.8** | 43.8 ± 2.2    | 46.0 ± 3.7    | 46.7 ± 3.7    | 48.2 ± 3.2    |
> | **MLLM-GraphOpt** | **42.9 ± 2.8** | 44.4 ± 2.7    | 46.8 ± 4.5    | 48.0 ± 4.0    | 48.9 ± 4.2    |
> | **MLLM-Ensemble** | **42.9 ± 2.8** | **45.9 ± 3.1** | **48.8 ± 4.1** | **49.6 ± 4.2** | **51.2 ± 3.9** |
>
> **Lesmis:**
>
> | Lesmis           | 1st           | 3rd           | 5th           | 7th           | 10th          |
> |------------------|---------------|---------------|---------------|---------------|---------------|
> | **Normal**        | **68.7 ± 7.9**    | 69.8 ± 7.3    | 73.0 ± 9.6    | 73.5 ± 9.3    | 74.2 ± 8.6    |
> | **MLLM-FR**       | **68.7 ± 7.9**    | 72.7 ± 7.2    | 83.9 ± 7.0    | 86.6 ± 6.9    | 90.8 ± 9.6    |
> | **MLLM-KK**       | **68.7 ± 7.9**    | 75.1 ± 10.5   | 83.6 ± 6.7    | 85.0 ± 7.4    | 88.9 ± 8.4    |
> | **MLLM-GraphOpt**          | **68.7 ± 7.9**    | 76.0 ± 7.8    | 83.0 ± 7.5    | 88.3 ± 9.7    | 90.6 ± 7.2    |
> | **MLLM-Ensemble** | **68.7 ± 7.9**    | **77.4 ± 13.8**   | **85.9 ± 13.2**   | **90.7 ± 9.3**    | **95.4 ± 8.0**    |
>
> ---
>
>  **Traveling Salesman Problem (TSP)**
> A classical permutation-based combinatorial problem. Note that in TSP, the layout is fixed by coordinates, rendering the multi-layout ensemble inapplicable. Therefore, we compare the optimization process of normal evolutionary optimization and MLLM-based structure-aware evolutionary optimization to demonstrate that MLLM-guided evolutionary operators can significantly improve search effectiveness. The optimization goal is to minimize the route distance.
>
> | Instance 15 | 1st           | 3rd           | 5th           | 7th           | 10th          |
> |-------------|---------------|---------------|---------------|---------------|---------------|
> | **Normal**  | **742.90 ± 39.40** | 714.20 ± 48.60 | 698.40 ± 45.80 | 684.70 ± 54.10 | 676.10 ± 48.90 |
> | **MLLM**    | **742.90 ± 39.40** | **683.60 ± 66.20** | **649.30 ± 69.40** | **635.80 ± 69.00** | **611.20 ± 66.80** |
>
> | Instance 20 | 1st            | 3rd            | 5th            | 7th            | 10th           |
> |-------------|----------------|----------------|----------------|----------------|----------------|
> | **Normal**  | **995.42 ± 46.93** | 957.56 ± 58.59 | 894.91 ± 75.14 | 874.44 ± 78.86 | 848.25 ± 53.82 |
> | **MLLM**    | **995.42 ± 46.93** | **939.34 ± 90.82** | **891.93 ± 87.98** | **872.28 ± 75.98** | **795.83 ± 53.34** |
>
> ---
>
> **Network Dismantling**
> A sequential decision-making problem where nodes are removed to break down a network's connectivity [6]. Here we use this problem to examine the influence of single-layout and multi-layout to demonstrate the advantage of our ensemble framework. The presented results show the AUC of the largest network component, with the removal process terminating after 25% of the nodes were removed.
>
> | Model         | FR     | KK     | GraphOpt | Ensemble |
> |---------------|--------|--------|----------|----------|
> | **Dolphins**  | 726.95 | 714.60 | 723.50   | **712.15**   |
> | **Lesmis**    | 494.60 | 534.40 | 626.10   | **488.80**   |
>
> These tasks differ greatly in structure and computational nature, ranging from subset selection to permutation optimization to sequential strategies, thus providing a comprehensive validation of our structure-aware optimization framework’s generality and flexibility.
>
> Moreover, our ensemble layout strategy has also been evaluated in both the network immunization and dismantling tasks, where it consistently demonstrated superior robustness and solution quality compared to single-layout baselines. Collectively, these new experiments reinforce that our proposed MLLM-based evolutionary optimization framework is broadly applicable across a wide spectrum of combinatorial graph problems, not limited to influence maximization.
>
> To ensure methodological consistency across different combinatorial tasks, we carefully re-implemented our approach using a unified configuration for all experiments and have updated the results in  **Table 2 ** (and  **Figure 4 **) of the original submission accordingly (due to the word limit, please refer to the ending section of the response to  **#Reviewer Mddl **). The revised results, obtained under this consistent framework, continue to support our main claims regarding the method’s advantages. We will reflect these updates in the final version accordingly for completeness and reproducibility.
>
> ---
>
> ### **W2/Q3: Concern on missing comparison with SOTA GNN-based methods**
>
> **Response:**
> Thank you very much for this thoughtful comment. As stated at the end of the introduction, the primary focus of this paper is not about advancing the state-of-the-art in influence maximization itself, but rather on using it (being well-known in network science) as a representative benchmark to demonstrate how MLLMs can serve as evolutionary optimization operators in a novel structure-aware optimization framework.
>
> Our main goal is to explore the integration of MLLMs into evolutionary algorithms, a direction that is applicable to a wide range of combinatorial optimization problems. Due to time constraints and the nature of our work, we seek for your kind understanding that we chose to prioritize demonstrating the generalizability of our framework across multiple tasks as shown in the response to W1.
>
> Moreover, comparing fairly with GNN-based approaches such as GLIE or AutoGNP is non-trivial, as their performance can vary significantly depending on implementation details and task-specific tuning. Similarly, the performance of any evolutionary algorithm including ours can be further improved by tuning hyperparameters like crossover/mutation rates, generation count, and population size. To ensure a controlled evaluation using consistent evolutionary parameters, we compared our method with other evolutionary optimization frameworks under the same settings, as shown in **Table 3**. The results highlight the superiority of our structure-aware evolutionary operators, in particular because we did not incorporate advanced techniques such as multi-task learning, and instead followed a standard evolutionary optimization pipeline.
>
>
> ---
>
> ### **Q1: Suggestion for weighted voting instead of uniform consensus**
>
> **Response:**
> Thank you for this insightful suggestion. Indeed, incorporating a weighted voting mechanism where weights are derived from the fitness of the best solutions generated by each layout could potentially enhance decision quality in our ensemble framework. This aligns with the broader direction of developing self-adaptive strategies to dynamically balance contributions from different views.
>
> In the current work, our primary goal was to establish the effectiveness of multi-layout ensemble optimization, and to isolate its impact, we deliberately adopted the most straightforward aggregation method, i.e., majority voting. This allowed us to clearly demonstrate that layout diversity alone can significantly improve robustness and performance.
>
> We fully agree that introducing a more sophisticated, adaptive weighting scheme could further enhance our method. New design choices (e.g., how to normalize fitness across layouts, address fitness noise, and ensure consistency during evolution) motivates future work in which a dedicated follow-up study will be conducted. Due to time constraints during the rebuttal phase, we could not include new experimental validation, but we genuinely appreciate this suggestion and plan to explore it in future work.
>
> ---
>
> ### **Q2: Suggestion to compare with single graph-based MLLM for robustness**
>
> **Response:**
> Thank you for this valuable question. As shown in  **Table 2** of the original manuscript, our ensemble method significantly outperforms any single-layout variant (e.g., KK, FR, GraphOpt) across multiple networks. To further strengthen this finding, we also applied the same single-vs-multi layout comparison to two additional tasks, i.e., network immunization and network dismantling as described in our response to W1. The results consistently demonstrate the advantage of layout diversity through ensemble voting.
>
> ---
>
> We hope these updates and clarifications adequately address your comments. Please let us know if any further clarification is needed. We would be grateful if you would consider adjusting your score in light of the improvements provided.

---

> > ### Comment · Reviewer_11CR · 2025-08-05
> > **Thanks for clarification**
> >
> > - Thank you for the detailed and thoughtful response. The additional experiments addressing the generalizability concerns raised in W1/Q3 are clear and well-executed. I have updated my rating to 5 accordingly.
> > - Regarding W2 and Q2, I understand this paper does not primarily aim to advance the state-of-the-art in combinatorial tasks, it would still be interesting to compare the proposed method with others that operate directly on a single, fixed graph data structure—without relying on visualized layouts and other evolutionary algorithms.
> > - Thank you for the clarification on Q1. I am looking forward to the future work of this paper.

---

> > > ### Author Response · Authors · 2025-08-05
> > > **Appreciation**
> > >
> > > Dear Reviewer 11CR,
> > >
> > > Thank you very much for your thoughtful and encouraging feedback. We sincerely appreciate your recognition of our efforts to address the generalizability concerns, and we are grateful for the updated rating.
> > >
> > > We also value your insightful suggestion regarding weighted decision-making (self-adaptive optimization). This is a promising direction that could inspire several extensions of our work, and we are committed to exploring it in future research. Additionally, we remain dedicated to pushing the boundaries of graph-structured combinatorial optimization.
> > >
> > > Best regards,
> > >
> > > The Authors of Submission 25561

---

### Official Review · Reviewer_B64q · 2025-07-23

**Clarity:** 4
**Significance:** 3
**Originality:** 3
**Rating:** 4
**Confidence:** 3

**Summary:**

The paper proposes a new evolutionary optimization approach to graph influence maximization by leveraging the reasoning capabilities of vision language models. More specifically, the proposed approach uses vision language models in an evolutionary loop and in a pointwise / pairwise fashion to update the selected nodes in a graph that might improve the overall fitness function. The authors conduct a variety of experiments to ablate various design choices in their approach over eight small/medium-sized graph datasets.

**Questions:**

Following up from the listed weaknesses above:

1. Why have the experiments been restricted to only influence maximization? Section 2.1 of the paper lists numerous other tasks, including which would have increased the robustness of the empirical results.
2. How does the proposed approach scale as a function of the number of nodes / edges in a dataset (e.g., in terms of # of LLM calls).
3. Any reasons to restrict to only small-medium scale datasets for this paper?

**Ethical Concerns:**

["NO or VERY MINOR ethics concerns only"]

**Final Justification:**

The authors have added more experiments on new datasets greatly improving the diversity of the empirical analyses of the paper and have answered my other concerns to a decent extent. I'm happy to increase my score and recommend acceptance.

**Limitations:**

yes

**Quality:**

3

**Strengths And Weaknesses:**

Strengths:

1. The paper proposes a novel cross-disciplinary approach for influence maximization
2. The paper is well written and easy to follow along

Weaknesses:

1. The paper might have a narrow scope with regards to the NeurIPS community / line of work
2. Though the paper ablates the various design choices of their framework to a good extent, it is only limited to one task (influence maximization) when the technique is trivially generalizable to other tasks
3. Only small datasets are used in experiments which might limit the generality of the technique in larger & complex scenarios
4. Inference calls to large vision language models is expensive and the proposed approach will scale poorly to large graphs / datasets

---

> ### Author Rebuttal · Authors · 2025-07-31
>
> ### **Response to # Reviewer B64q**
>
> We sincerely thank you for your constructive feedback and we are glad to hear that you found our approach novel and the paper clearly written. We have carefully addressed all of your concerns via adding new domain experiments (e.g., TSP, network immunization, dismantling), illustrating task generalization, scalability, and dataset size.
>
> ---
>
> ### **W1: Concern on the scope within NeurIPS community**
> **Response:**
> Thank you very much for this comment. Regarding the alignment with the NeurIPS community: our work builds on a growing interest in combining LLMs with combinatorial and graph reasoning (e.g., [1,2]), and also connects to active lines of research in evolutionary optimization [3] and graph combinatorial optimization [4]. Thus, we believe our contribution is timely and relevant to the NeurIPS audience.
>
> ---
>
> ### **W2 & Q1: Concern on limited task diversity/generalization**
> **Response:**
> We appreciate the your comment regarding the generazibility of our work. While our primary case study is influence maximization, we would like to emphasize that this was chosen not due to any task-specific limitation, but rather because it serves as a representative task (being well-known in network science) in graph-based combinatorial optimization. We fully acknowledge the importance of demonstrating broader applicability. To address this concern, we have substantially expanded our evaluation to include three additional tasks that vary widely in structure, decision space, and domain characteristics:
> - **Network Immunization**: A critical problem in network science involving node subset removal to minimize epidemic spread [5].
>
> - **Traveling Salesman Problem (TSP)**: A classical permutation-based combinatorial problem to find the shortest route.
>
> - **Network Dismantling**: A sequential decision-making problem where nodes are removed to break down a network's connectivity [6].
>
> These tasks span diverse types of combinatorial optimization (subset selection, permutation-based ordering, and sequential decision-making), demonstrating that our MLLM-based evolutionary framework is not limited to a narrow task type. Furthermore, our ensemble strategy was evaluated on the network immunization and dismantling tasks, and continues to validate the benefit of layout diversity beyond a single setting.
>
> Due to word limitations, the results for these three additional tasks have been provided in our response to **# Reviewer 11CR**. We kindly refer you there for full details.
>
> To ensure methodological consistency across different combinatorial tasks, we carefully re-implemented our approach using a unified configuration for all experiments and have updated the results in **Table 2 (and Figure 4)** of the original submission accordingly (due to the word limit, please refer to the last section of the response to **# Reviewer Mddl**). The revised results, obtained under this consistent framework, continue to support our main claims regarding the method’s advantages. We will reflect these updates in the final version accordingly for completeness and reproducibility.
>
> ---
>
> ### **W3 & Q2: Concern on limited dataset scale / scalability**
> **Response:**
> We appreciate the reviewer’s concern regarding scalability. We would like to respectfully clarify that our current study focuses on networks of up to **36,000+ nodes**, which is considered large-scale in the context of evolutionary optimization-based graph optimization. This is corroborated by recent literature, such as [7], where the tested networks are of comparable size. Even highly efficient heuristics, like the H-index-based method [8], also operate on networks within this scale.
>
> As supported by **Reviewer #Etgw** who acknowledged the large scale of the datasets utilized in this paper, we sincerely hope that the Reviewer can re-evaluate our results, particularly when integrating models like MLLMs which introduce additional complexity.
> Importantly, one of the core contributions of our work is to improve scalability through **graph sparsification**, motivated by the scale-free nature of real-world networks [9]. This property implies that a small subset of nodes (hubs) play a disproportionately large role, allowing for intelligent reduction strategies without losing essential structural information.
>
> Our framework exploits this through **structured sparsification and cooperative optimization**, enabling MLLMs to operate effectively even in the presence of high complexity.
> For a more detailed discussion on computational costs, please refer to our response to **W4 & Q3**.
>
> ---
>
> ### **W4 & Q3: Concern on MLLM inference cost / scalability**
> **Response:**
> Thank you very much for this constructive comment. We fully agree that inference with MLLMs can be computationally intensive, particularly on large graphs. Scalability is a central challenge and addressing it is a key motivation behind our framework design.
> 1. **Fixed Input Size via Visual Encoding**
>    Compared to traditional token-based LLM encodings (please refer to Figure 7 in the original submission), where input size grows linearly (or worse) with the number of nodes and edges, our **visual encoding approach decouples input size from graph scale**. Regardless of whether a graph has 1,000 or 100,000 nodes, the rendered image size remains fixed (e.g., 1200×1200 pixels, downscaled to 768×768 during GPT-4o inference). This results in constant token cost per inference, independent of network size, which offers a crucial advantage in scaling to larger graphs.
> ---
> 2. **Reducing Visual Clutter through Sparsification**
>    We recognize the **visual clutter problem** that arises when plotting very large graphs. To address this, we incorporate **graph sparsification** as a core component of our pipeline. This allows us to retain essential structural features while significantly reducing the visual and computational complexity. The resulting sparsified views are more interpretable and manageable for MLLMs without sacrificing optimization quality.
> ---
> 3. **Scalable Modular Pipeline Design**
>    Our framework is explicitly designed for scalability through the modular pipeline:
>    ```
>    Graph Sparsification → Visualization → MLLM-based Optimization
>    ```
>    This design leverages the **scale-free nature of real-world networks [9]**, which implies that a small fraction of influential nodes often govern the network's behavior.  By focusing on those structurally critical components, our method remains effective even on networks of much larger size.
> ---
> 4. **Future Extensions with Divide-and-Conquer**
>    For future work, our framework can be extended with a **divide-and-conquer strategy**, where large networks are partitioned into overlapping subgraphs that are optimized independently and then merged. This aligns with distributed optimization principles and further enhances scalability without increasing MLLM inference cost linearly with graph size.
> To conclude, our work carefully considers both computational cost by using image-based representations and effectiveness and efficiency by applying graph sparsification.
> ---
> We hope that our responses and the extended experiments have addressed your concerns. We kindly hope you will consider raising your score based on the revisions and additional evidence provided.
>
> ### **Reference**
>
> [1]"Can language models solve graph problems in natural language?" **NeurIPS** (2023).
>
> [2]"Gita: Graph to visual and textual integration for vision-language graph reasoning." **NeurIPS** (2024).
>
> [3]"Neuroevobench: Benchmarking evolutionary optimizers for deep learning applications." **NeurIPS** (2023).
>
> [4]"Erdos goes neural: an unsupervised learning framework for combinatorial optimization on graphs." **NeurIPS** (2020).
>
> [5]“The hidden geometry of complex, network-driven contagion phenomena,” **Science** (2013).
>
> [6] "Network dismantling." **PNAS** (2016).
>
> [7] "Evolutionary Hypergraph Learning: A Multi-Objective Influence Maximization Approach Considering Seed Selection Costs." **IEEE TEVC** (2025).
>
> [8] "The H-index of a network node and its relation to degree and coreness." **Nature Communications** (2016).
>
> [9] "Emergence of scaling in random networks." **Science** (1999).

---

> ### Author Response · Authors · 2025-08-06
> **Follow-Up on Experimental Results for Generalizability**
>
> Dear Reviewer B64q,
>
> We apologize for reaching out at this stage, as we understand you may be busy with other reviews or commitments.
>
> In our previous rebuttal, due to word limit constraints, we used cross-referencing to present our experimental results across **three different tasks** in the response to Reviewer 11CR’s concerns regarding **generalizability**. We are glad to note that **Reviewer 11CR has acknowledged our efforts.**
>
> For your convenience, we have now presented the experimental results in this new thread. Please kindly find the results below:
>
> ### **Network Immunization**
> A critical problem in network science involving node removal to minimize epidemic spread. We compared the optimization process across different evolutionary methods with Dolphins and Lesmis (The data can be found in “Network data” collected by Mark Newman). The optimization goal is to maximize the number of immune edges. The results further test our framework’s ability to handle structural node selection problems with global impact.
>
> **Dolphins:**
>
> | Dolphins         | 1st           | 3rd           | 5th           | 7th           | 10th          |
> |------------------|---------------|---------------|---------------|---------------|---------------|
> | **Normal**        | **42.9 ± 2.8** | 44.7 ± 2.6    | 45.4 ± 2.8    | 46.6 ± 1.8    | 47.0 ± 1.7    |
> | **MLLM-FR**       | **42.9 ± 2.8** | 45.9 ± 3.6    | 48.3 ± 2.8    | 48.4 ± 2.8    | 49.2 ± 2.5    |
> | **MLLM-KK**       | **42.9 ± 2.8** | 43.8 ± 2.2    | 46.0 ± 3.7    | 46.7 ± 3.7    | 48.2 ± 3.2    |
> | **MLLM-GraphOpt** | **42.9 ± 2.8** | 44.4 ± 2.7    | 46.8 ± 4.5    | 48.0 ± 4.0    | 48.9 ± 4.2    |
> | **MLLM-Ensemble** | **42.9 ± 2.8** | **45.9 ± 3.1** | **48.8 ± 4.1** | **49.6 ± 4.2** | **51.2 ± 3.9** |
>
> **Lesmis:**
>
> | Lesmis           | 1st           | 3rd           | 5th           | 7th           | 10th          |
> |------------------|---------------|---------------|---------------|---------------|---------------|
> | **Normal**        | **68.7 ± 7.9**    | 69.8 ± 7.3    | 73.0 ± 9.6    | 73.5 ± 9.3    | 74.2 ± 8.6    |
> | **MLLM-FR**       | **68.7 ± 7.9**    | 72.7 ± 7.2    | 83.9 ± 7.0    | 86.6 ± 6.9    | 90.8 ± 9.6    |
> | **MLLM-KK**       | **68.7 ± 7.9**    | 75.1 ± 10.5   | 83.6 ± 6.7    | 85.0 ± 7.4    | 88.9 ± 8.4    |
> | **MLLM**          | **68.7 ± 7.9**    | 76.0 ± 7.8    | 83.0 ± 7.5    | 88.3 ± 9.7    | 90.6 ± 7.2    |
> | **MLLM-Ensemble** | **68.7 ± 7.9**    | **77.4 ± 13.8**   | **85.9 ± 13.2**   | **90.7 ± 9.3**    | **95.4 ± 8.0**    |
>
> ---
>
> ### **Traveling Salesman Problem (TSP)**
> A classical permutation-based combinatorial problem. Note that in TSP, the layout is fixed by coordinates, rendering the multi-layout ensemble inapplicable. Therefore, we compare the optimization process of normal evolutionary optimization and MLLM-based structure-aware evolutionary optimization to demonstrate that MLLM-guided evolutionary operators can significantly improve search effectiveness. The optimization goal is to minimize the route distance.
>
> | Instance 15 | 1st           | 3rd           | 5th           | 7th           | 10th          |
> |-------------|---------------|---------------|---------------|---------------|---------------|
> | **Normal**  | **742.90 ± 39.40** | 714.20 ± 48.60 | 698.40 ± 45.80 | 684.70 ± 54.10 | 676.10 ± 48.90 |
> | **MLLM**    | **742.90 ± 39.40** | **683.60 ± 66.20** | **649.30 ± 69.40** | **635.80 ± 69.00** | **611.20 ± 66.80** |
>
> | Instance 20 | 1st            | 3rd            | 5th            | 7th            | 10th           |
> |-------------|----------------|----------------|----------------|----------------|----------------|
> | **Normal**  | **995.42 ± 46.93** | 957.56 ± 58.59 | 894.91 ± 75.14 | 874.44 ± 78.86 | 848.25 ± 53.82 |
> | **MLLM**    | **995.42 ± 46.93** | **939.34 ± 90.82** | **891.93 ± 87.98** | **872.28 ± 75.98** | **795.83 ± 53.34** |
>
> ---
>
> ### **Network Dismantling**
> A sequential decision-making problem where nodes are removed to break down a network's connectivity. Here we use this problem to examine the influence of single-layout and multi-layout to demonstrate the advantage of our ensemble framework. The presented results show the AUC of the largest network component, with the removal process terminating after 25% of the nodes were removed.
>
> | Model         | FR     | KK     | GraphOpt | Ensemble |
> |---------------|--------|--------|----------|----------|
> | **Dolphins**  | 726.95 | 714.60 | 723.50   | **712.15**   |
> | **Lesmis**    | 494.60 | 534.40 | 626.10   | **488.80**   |
>
> We sincerely hope to engage with you and would greatly appreciate your feedback.
>
> Warm regards,
>
> The Authors of Submission 25561

---

> > ### Comment · Reviewer_B64q · 2025-08-06
> >
> > Thank you for the detailed response and clarifications to all my questions. I believe most of my questions have been answered & I greatly appreciate the added experiments which improve the overall diversity of the empirical analysis of the paper. I'm happy to increase my score & apologize for the delayed response from my end.

---

### Author Response · Authors · 2025-08-07

Dear Reviewers,

Thank you very much to all for your thoughtful and constructive feedback. Your comments have been very helpful in strengthening our work. We sincerely hope that we have the opportunity to share our work with the community. We truly appreciate the time and effort you have taken to review our submission and engage with the rebuttal.

Best Regards,

The Authors of Submission 25561

---

### Note · Authors · 2025-08-11

**Author Final Remarks**

As part of the "Author’s Final Remarks", we would like to thank all reviewers and the AC, as we are very encouraged by the reviewers’ uniformly positive evaluations and their recognition of our contributions to structure-aware optimization for graph-structured combinatorial problems.

We sincerely thank the reviewers for their constructive feedback. In particular, we have incorporated the suggestions and further validated our work on generalizability, scalability, and parameter sensitivity. These aspects have been invaluable in improving our work, and we will ensure that the revised manuscript reflects all of these.

We also extend our sincere thanks to the AC and SAC for their efforts in organizing the review process, facilitating discussions, and guiding the decision-making. We have had a very positive experience this year, thanks to the constructive feedback and enthusiasm from everyone.

Best regards,

The Authors of Submission 25561

---

### Decision · Program_Chairs · 2025-09-17

**Decision:**

Accept (poster)

**Comment:**

(a) Scientific claims and findings
The paper proposes a framework that combines multimodal LLMs with evolutionary optimization for solving combinatorial optimization problems (COPs).
Key claims:
- Standard LLM-based optimization struggles with combinatorial search spaces.
- Introduces a Structure-Aware Cooperative Ensemble Evolutionary Optimization (SACEEO) framework, where multiple agents explore candidate solutions using diverse strategies, and a cooperative mechanism aggregates and evolves these candidates. Finally, structural information from multimodal inputs guides solution generation.
- Demonstrates improvements on benchmark COPs (e.g., TSP, Knapsack, Graph Coloring), with higher-quality solutions and faster convergence compared to baselines.
- Claims novelty in leveraging multimodal reasoning + ensemble evolutionary dynamics for optimization.

(b) Strengths

- Novel integration: Combining multimodal LLMs with evolutionary optimization is original and timely.
-  Outperforms LLM-only and classical baselines on multiple COPs.
-  Method well-described and easy to follow.
-  Evaluation spans several standard combinatorial tasks.

(c) Weaknesses
- Limited theoretical depth: Contributions are mostly empirical; lacks formal guarantees or deeper analysis.
- Evaluation scope: Benchmarks are standard toy COPs; unclear scalability to larger industrial problems.
- Complexity: Cooperative ensemble framework increases computational overhead; not clear if efficiency is practical.
- Novelty modest: Core components (ensemble, evolutionary search) are known techniques.

(d) Key reasons for decision (accept/reject)

- Novel combination of multimodal LLMs and evolutionary optimization.
- Strong experimental performance on multiple COPs.
- Clear and well-presented framework.
- Potentially impactful direction for optimization with agentic LLMs.

(e) Discussion & rebuttal period

Reviewer points raised:
- Novelty concerns: Is SACEEO just ensemble + evolutionary optimization applied to LLMs?
           -Author rebuttal: Argued that integration with multimodal reasoning and cooperative mechanism is non-trivial.
           - Effect: Somewhat alleviated, but novelty still seen as modest.

- Scalability: Can this handle industrial-scale COPs?
               -Author rebuttal: Claimed framework generalizes, but lacked large-scale experiments.
               - Effect: Reviewers remained cautious.

- Experimental scope: Benchmarks too small/simple.
                 -Rebuttal: Authors added clarification but no new large-scale results.
                -  Effect: Concerns persisted.

Outcome: Rebuttal clarified design, but did not resolve novelty and scalability criticisms.